# GAP PRESERVING DISTILLATION BY BUILDING BIDIRECTIONAL MAPPINGS WITH A DYNAMIC TEACHER

**Yong Guo**$^{1*}$, **Shulian Zhang**$^{2*}$, **Haolin Pan**$^2$, **Jing Liu**$^3$, **Yulun Zhang**$^4$, **Jian Chen**$^{2\dagger}$
$^1$Max Planck Institute for Informatics,  $^2$South China University of Technology,
$^3$Monash University,  $^4$Shanghai Jiao Tong University
guoyongcs@gmail.com, {seshulian,sephl}@mail.scut.edu.cn,
jing.liu1@monash.edu, yulun100@gmail.com, ellachen@scut.edu.cn

## ABSTRACT

Knowledge distillation aims to transfer knowledge from a large teacher model to a compact student counterpart, often coming with a significant performance gap between them. Interestingly, we find that a too-large performance gap can hamper the training process. To alleviate this, we propose a **Gap Preserving Distillation (GPD)** method that trains an additional dynamic teacher model from scratch along with the student to maintain a reasonable performance gap. To further strengthen distillation, we develop a hard strategy by enforcing both models to share parameters. Besides, we also build the soft bidirectional mappings between them through *Inverse Reparameterization (IR)* and *Channel-Branch Reparameterization (CBR)*. IR initializes a larger dynamic teacher with approximately the same accuracy as the student to avoid a too large gap in early stage of training. CBR enables direct extraction of an effective student model from the dynamic teacher without post-training. In experiments, GPD significantly outperforms existing distillation methods on top of both CNNs and transformers, achieving up to 1.58% accuracy improvement. Interestingly, GPD also generalizes well to the scenarios without a pre-trained teacher, including training from scratch and fine-tuning, yielding a large improvement of 1.80% and 0.89% on ResNet18, respectively. The code is available at `https://github.com/guoyongcs/GPD`.

## 1 INTRODUCTION

Deep neural networks have achieved remarkable success across various domains (Gu & Dao, 2024; Liu et al., 2023b; Touvron et al., 2023; Wang et al., 2023; Zhang et al., 2023a). However, the high accuracy of these models often comes at the cost of large model sizes. Recent studies have begun to explore methods to reduce model complexity. In parallel to model compression techniques (Liu et al., 2023a; Wei et al., 2022; Hu et al., 2024; Xiao et al., 2023), knowledge distillation (KD) (Hinton et al., 2015) offers a solution by transferring knowledge from complex, high-capacity models to simpler, more lightweight models, thereby achieving effective model compression.

The standard practice of KD is to train a smaller student model to mimic the behavior or predictions of a teacher model (Sun et al., 2024; Jin et al., 2023; Zhao et al., 2022a; Li et al., 2023; Zhao et al., 2022a; Passalis et al., 2021; Tian et al., 2020; Zagoruyko & Komodakis, 2017a; Heo et al., 2019a; Chen et al., 2021b; Yang et al., 2021; Peng et al., 2019). However, existing methods often exploit a fixed pre-trained teacher model but it does not always guide the training of student in the most effective way. To be specific, it has been shown that it is often non-trivial for the student to obtain promising knowledge/improvements from the teacher when there is a very large gap between the teacher and student models, especially in the early training stage. In contrast, a weaker teacher, together with a smaller performance gap from the student, has been shown to be a better choice (Son et al., 2021; Yang et al., 2019b; Mirzadeh et al., 2020; Wang et al., 2022; Gao et al., 2021). In fact, it can be theoretically proved by (Wang et al., 2022) that weak models "have higher mutual information regarding the input" compared to stronger teacher models, which can enhance knowledge

---

$^*$Equal contribution.
$^\dagger$Corresponding author.

distillation. Interestingly, this phenomenon can be intuitively understood, just like it is hard for a child to learn advanced mathematics directly from a senior professor. Instead, it would be better to teach him/her to count or learn elementary mathematics first. In other words, for distillation, a student model should learn from a teacher that has an appropriate knowledge/performance gap over it. Nevertheless, even with a relatively weak teacher that has a small performance gap, a fixed model that is commonly used would eventually become useless since the student will gradually improve and surpass the teacher at some time. Thus, how to adaptively control the performance gap within a reasonable range throughout the whole distillation process is a critical problem.

To address this, we seek to explicitly control the performance gap to boost knowledge transfer. However, it is non-trivial to determine the best range of the performance gap, since this gap may vary significantly across different datasets and tasks. For example, a performance gap of 3% or 5% could be good for training classification models on ImageNet. But this gap will definitely not be a suitable value for training on MNIST since almost all the models have an accuracy of over 98%. More critically, if we consider other tasks, e.g. image restoration, accuracy cannot be directly used to measure the gap. Instead of seeking a universal "best gap", we propose to cast the problem of determining the best gap into building a suitable gap in terms of model size between the student and the teacher. We propose to introduce a learnable dynamic teacher (DT) model besides the pre-trained static teacher model, training it from scratch together with the student. Since DT is a larger model and often has higher accuracy than the student, we are able to keep a promising performance gap between DT and student during training. In addition, we hope to build a stronger connection between DT and student to transfer knowledge in a more explicit way, in order to further enhance the performance. To achieve this, we develop a hard strategy for distillation that forces DT and the student to share their parameters and encourage parameter inheritance. In addition to hard strategy, we also build a soft bidirectional mapping between them via a novel reparameterization method.

In this paper, we make the following key contributions: **1)** We propose a **Gap Preserving Distillation (GPD)** method that enhances distillation performance by introducing an additional dynamic teacher (DT). We simultaneously train DT and the student to maintain a reasonable accuracy gap between them during the whole training process. In this way, it becomes possible to build a dynamic performance gap *at every iteration* between the student and teacher, which is essentially different from existing work. **2)** We develop a **hard strategy** for distillation where the student and the teacher share the same set of parameters. The key idea is that, since the teacher is often easier to obtain high accuracy, it may also be easier for the student to get promising improvement if it directly inherits well-learned parameters from the teacher, as empirically shown in Table 4. **3)** We explicitly enhance knowledge transfer by building *bidirectional mappings* between DT and the student via **Inverse Reparameterization (IR)** and **Channel-Branch Reparameterization (CBR)**. IR constructs the dynamic teacher model by expanding the student model with an arbitrary expansion ratio along both the channel and branch dimensions, while preserving the same accuracy as the student model. This guarantees that both DT and the student can start from the same initial point and thus avoid a too-large performance gap in early the training stage. Interestingly, IR is not designed just for distillation, but a general initialization method for building any-size pre-trained models, yielding additional contributions to the community beyond distillation. On the other hand, our CBR seeks to extract an effective student model from the shared parameters with DT without any post-training. Unlike existing reparameterization approaches, CBR does not equivalently transform a model into a more compact one, but extracts an effective student out of DT. **4)** In experiments, GPD consistently outperforms existing distillation methods on top of both CNN and transformer architectures. We emphasize that GPD is very flexible in that it also generalizes well to other training settings, including both training from scratch and fine-tuning, which is rarely reported by other methods.

## 2 RELATED WORK

**Knowledge distillation.** Knowledge distillation (KD) (Hinton et al., 2015) transfers knowledge from a teacher to a smaller student model. Methods improve this by focusing on logits or intermediate features (Sun et al., 2024; Jin et al., 2023; Zhao et al., 2022a; Li et al., 2023; Passalis et al., 2021; Tian et al., 2020; Zagoruyko & Komodakis, 2017a; Heo et al., 2019a; Chen et al., 2021b; Heo et al., 2019b; Kim et al., 2018). Standard methods prioritize fully converged teachers with high performance, yet the performance gap can hinder knowledge transfer (Wang et al., 2022; Cho & Hariharan, 2019; Yuan et al., 2019; Gao et al., 2021). Strategies to address this in-

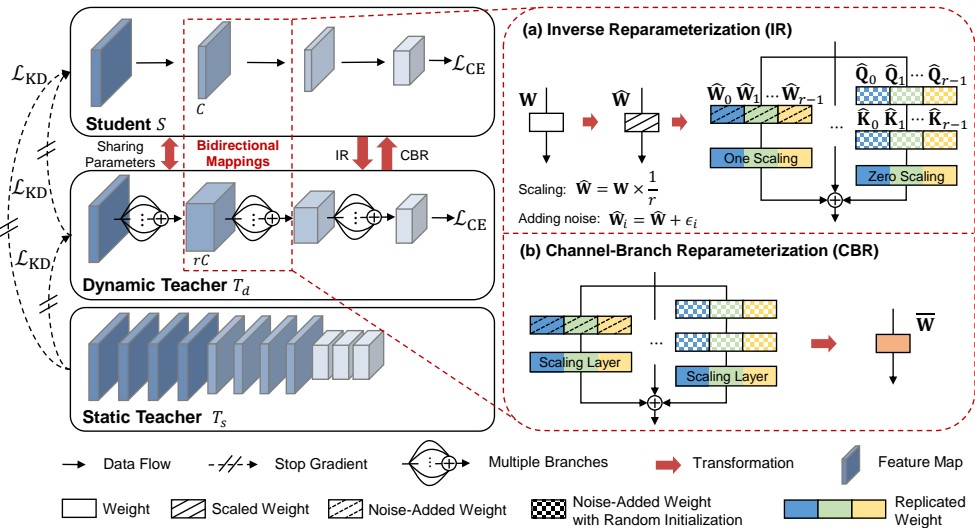

Figure 1: Overview of the proposed Gap Preserving Distillation (GPD) method. Besides the static teacher, we introduce an additional dynamic teacher and train it from scratch along with the student. The student model shares parameters with the dynamic teacher via *Inverse Reparameterization (IR)* and *Channel-Branch Reparameterization (CBR)*. (a) The dynamic teacher is constructed through IR (top right) from the student model. For any layer, we replicate the weights along the channel dimension to build a wider layer while introducing additional branches to construct a multi-branch architecture. In order to maintain the same accuracy as the student, we only activate the first branch that contains the original student weights and zero out all the other extra branches, *i.e.*, one-scaling and zero-scaling. (b) We extract a promising student from the dynamic teacher via CBR. The expanded multi-branch architecture can be merged into the student's single-branch architecture using a similar way proposed by OREPA (Hu et al., 2022). After that, given an expansion ratio $r$, we directly extract the first $1/r$ parameters multiplied by a scaling factor (see details in Section 3.3.1).

clude using intermediate-stage teachers (Cho & Hariharan, 2019; Zhao et al., 2022b), pre-training student-friendly teacher model (Yang et al., 2019a; Park et al., 2021; Dong et al., 2024), introducing intermediate-sized assistant teachers (Mirzadeh et al., 2020; Son et al., 2021) or introducing auxiliary networks (Gao et al., 2021). These methods often rely on specially designed and pre-trained intermediate models. Feature-based methods like DTSKD (Li et al., 2024b) and DiffKD (Huang et al., 2023) focus on bridging semantic gaps or denoising features. SCKD (Zhu & Wang, 2021) optimizes transfer using gradient similarity. Recent works refine soft labels (Yuan et al., 2024; Rao et al., 2023) or student's output entropy (Zhu et al., 2024a) to enhance knowledge transfer. In contrast, our GPD constructs a trainable dynamic teacher based on the student model, maintaining an appropriate accuracy gap throughout distillation for effective knowledge transfer.

**Reparameterization.** Structural reparameterization (Ding et al., 2021a;b) has gained attention in tasks such as compact model design (Dosovitskiy et al., 2021), architecture search (Chen et al., 2019; Zhang et al., 2023c), and pruning (Ding et al., 2020). RepVGG (Ding et al., 2021b) transforms training-time structures into equivalent, simpler inference structures. Other methods like DiracNet (Zagoruyko & Komodakis, 2017b), ACB(Ding et al., 2019), DO-Conv(Cao et al., 2022), and ExpandNet (Guo et al., 2020) also achieve structural reparameterization. OREPA (Hu et al., 2022) reduces training costs by online reparameterization. In contrast, our CBR enables the extraction of an effective student model from the dynamic teacher, enhancing knowledge transfer to the compact student model.

**Model expansion.** Net2Net (Chen et al., 2016) pioneered functional-preserving model expansion. bert2BERT (Chen et al., 2021a) applied this to Transformers, with other works focusing on depth growth (Dong et al., 2020; Chang et al., 2018; Yang et al., 2020; Gong et al., 2019). Staged Training (Shen et al., 2022) and LEMON (Wang et al., 2024) further expanded both width and depth. ControlNet (Zhang et al., 2023b) duplicates the model and adds zero convolution layers to maintain equivalence, freezing the original parameters for fine-tuning. ControlNet (Zhang et al., 2023b) duplicates the model, adds zero convolution layers, and freezes original parameters for fine-tuning. In contrast, our GPD expands the model and enables adaptive switching between compact and expanded models during training, maintaining an appropriate performance gap.

## 3 GAP PRESERVING DISTILLATION

In this work, we develop a **Gap Preserving Distillation (GPD)** method that enhances knowledge distillation by preserving an appropriate performance gap between the teacher and student throughout the whole distillation process. Instead of directly defining how large the gap should be, we propose to learn an additional dynamic teacher (DT) model as a proxy to maintain this gap, which is more flexible and controllable. Unlike existing methods, we build bidirectional mappings between DT and student to strengthen their connections. These two mappings can be achieved by the Inverse Reparameterization (IR) method and the Channel-Branch Reparameterization (CRB) method, respectively. The overview of our method is shown in Figure 1 and Algorithm 1.

### 3.1 DISTILLATION WITH DYNAMIC TEACHER

Popular KD methods often exploit a static teacher $T_s$ to guide the training of the student $S$ (Sun et al., 2024; Jin et al., 2023; Zhao et al., 2022a; Li et al., 2023; Zhao et al., 2022a; Furlanello et al., 2018; Zhao et al., 2022a; Romero et al., 2015; Passalis et al., 2021; Tian et al., 2020; Zagoruyko & Komodakis, 2017a; Heo et al., 2019a; Chen et al., 2021b; Heo et al., 2019b; Kim et al., 2018). Unlike them, our Gap Preserving Distillation (GPD) introduces a learnable dynamic teacher (DT) model $T_d$, as shown in Figure 1. Moreover, we seek to boost knowledge transfer by not only optimizing the distillation objectives, but also by inheriting parameters from a better teacher model. To achieve this, we exploit parameter sharing techniques (Zhao et al., 2023; Xie et al., 2022; Zhang et al., 2022; Ma et al., 2022; Zhou et al., 2022) and enforce $T_d$ and $S$ to share the same set of parameters, which turns out to be particularly effective (see effectiveness in Section 5).

As for training, besides the standard objective of KD methods, we introduce an additional loss related to DT $\mathcal{L}_{\text{GPD}}$. Let $\mathcal{L}_{\text{CE}}$ be the cross-entropy loss, and $S(x)$ denotes the student model's prediction based on input $x$. We use $\psi(\cdot)$ to represent a function that extracts desired knowledge from models, which could be either logits or features. $\mathcal{L}_{\text{KD}}$ measures the discrepancy between the knowledge of two models. Given an image-label pair $(x, y)$, the objective function of GPD with a dynamic teacher is formulated as follows:

$$\mathcal{L}_{\text{total}} = \underbrace{\mathcal{L}_{\text{CE}}(S(x), y) + \mathcal{L}_{\text{KD}}(\psi(S(x)), \psi(T_s(x)))}_{\text{standard objective of KD methods}} + \mathcal{L}_{\text{GPD}}. \tag{1}$$

As for $\mathcal{L}_{\text{GPD}}$, in Figure 1, we seek to use $T_d$ to guide the training of $S$ and thus introduce a KD loss $\mathcal{L}_{\text{KD}}(\psi(S(x)), \psi(T_d(x)))$. Regarding the training of $T_d$, we minimize both the cross-entropy loss $\mathcal{L}_{\text{CE}}$ and a KD loss between $T_d$ and $T_s$ via $\mathcal{L}_{\text{KD}}(\psi(T_d(x)), \psi(T_s(x)))$. Thus, $\mathcal{L}_{\text{GPD}}$ becomes

$$\mathcal{L}_{\text{GPD}} = \mathcal{L}_{\text{CE}}(T_d(x), y) + \lambda\mathcal{L}_{\text{KD}}(\psi(S(x)), \psi(T_d(x))) + \mathcal{L}_{\text{KD}}(\psi(T_d(x)), \psi(T_s(x))). \tag{2}$$

Where $\lambda$ controls the importance of the distillation loss between the student and DT. We set $\lambda = 3$ in all experiments. To avoid learning knowledge from a weaker model, we consider single-way knowledge transfer and apply stop-gradient on all the KD losses.

### 3.2 BUILD DYNAMIC TEACHER VIA INVERSE REPARAMETERIZATION

In order to build a suitable dynamic teacher, we develop an Inverse Reparameterization (IR) method to build a larger model from the student with any expansion ratio $r$, see Figure 1 (top right). To achieve this, we expand the model along both the channel and branch dimensions. One key characteristic is that the expanded dynamic teacher shares approximately the same accuracy as the student. Notably, IR can serve as a general initialization method for building any-size pre-trained models, yielding additional contributions to the community beyond distillation.

#### 3.2.1 CHANNEL-LEVEL INVERSE REPARAMETERIZATION

In the channel level, we seek to expand the number of channels in each layer by an expansion ratio $r$ without sacrificing accuracy. To this end, we propose a channel-level inverse reparameterization strategy, as shown in Figure 2. The key idea is to replicate the weights of the student and introduce a scale factor to compensate for the increased number of channels.

---

**Algorithm 1** Training process of Gap Preserving Distillation (GPD).

---

**Input:** Student $S$, static teacher $T_s$, epochs $N$, step size $\eta$, model parameters $\mathbf{W}$, weight of standard knowledge distillation loss $\lambda$, knowledge function $\psi(\cdot)$, training data $(x, y)$

1: Obtain dynamic teacher $T_d$: $\mathbf{W}_d \leftarrow \text{IR}(\mathbf{W})$ // Inverse Reparameterization
2: **for** $i = 1$ to $N$ **do**
3:      Forward propagation using the dynamic teacher via $\hat{y}_d = T_d(x)$;
4:      Compute gradients for dynamic teacher:
5:          $\mathbf{G}_d \leftarrow \nabla_{\mathbf{W}_d} \mathcal{L}_{\text{CE}}(\hat{y}_d, y) + \nabla_{\mathbf{W}_d} \mathcal{L}_{\text{KD}}(\psi(\hat{y}_d), \psi(T_s(x)))$;
6:      Obtain student $S$ from dynamic teacher $T_d$:
7:          $\mathbf{W}_s \leftarrow \text{CBR}(\mathbf{W}_d)$ // Channel-Branch Reparameterization
8:      Forward propagation using the student via $\hat{y}_s = S(x)$;
9:      Compute gradients for student:
10:     $\mathbf{G}_s \leftarrow \nabla_{\mathbf{W}_s} \mathcal{L}_{\text{CE}}(\hat{y}_s, y) + \lambda \nabla_{\mathbf{W}_s} \mathcal{L}_{\text{KD}}(\psi(\hat{y}_s), \psi(T_s(x)))$
11:            $+ \nabla_{\mathbf{W}_s} \mathcal{L}_{\text{KD}}(\psi(\hat{y}_s), \psi(T_d(x)))$;
12:     Update parameters $\mathbf{W}_d$ sharing between student $S$ and dynamic teacher $T_d$:
13:          $\mathbf{W}_d \leftarrow \mathbf{W}_d - \eta(\mathbf{G}_d + \mathbf{G}_s)$;
14: **end for**

---

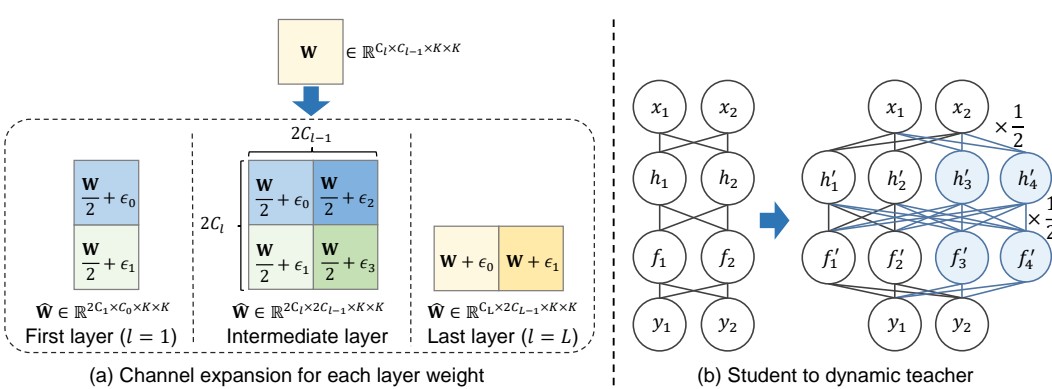

(a) Channel expansion for each layer weight        (b) Student to dynamic teacher

Figure 2: Illustration of channel-level inverse reparameterization with an expansion ratio of 2. (a) For the first layer, weights are scaled by 2 and replicated along the output channel dimension, expanding from $C_1 \times C_0$ to $2C_1 \times C_0$. For intermediate layers, weights are scaled by 2, then replicated along both input and output dimensions, expanding from $C_l \times C_{l-1}$ to $2C_l \times 2C_{l-1}$. For the last layer, weights are replicated along the input dimension, expanding from $C_L \times C_{L-1}$ to $C_L \times 2C_{L-1}$. (b) Inverse re-parameterizing the student model (left) to construct the dynamic teacher model (right) by expanding channels from 2 to 4 following the procedures exemplified in (a), while preserving the initial input-output mapping.

Considering a student model with $L$ convolutional layers, let $\mathbf{W}^l \in \mathbb{R}^{C_l \times C_{l-1} \times K \times K}$ be the weight of the $l$-th layer. Here, $C_l$ and $C_{l-1}$ represent the number of output and input channels, and $K \times K$ denotes the kernel size. Let $\widehat{\mathbf{W}}^l \in \mathbb{R}^{rC_l \times rC_{l-1} \times K \times K}$ be the expanded weight matrix for layer $l$ of the dynamic teacher, given the expansion ratio $r$. To better illustrate our method, we divide all layers into three groups, including the first layer, intermediate layers, and the last layer. In Figure 2, we take the expansion ratio $r = 2$ for example to illustrate our method. For the first layer ($l = 1$), we replicate $\mathbf{W}^1$ by $r$ times and scale them by $1/r$. In this way, given the same input $x$, the output would be $1/r$ of the original one. Nevertheless, since the number of output channels has been extended from $C_1$ to $rC_1$, summing up all the channels would obtain the same value as the original output. For convenience, we use $\widehat{\mathbf{W}}^1 \in \mathbb{R}^{rC_1 \times C_0 \times K \times K}$ to denote the weight scaled by $1/r$. For the last layer ($l = L$), the original $\mathbf{W}^L$ is replicated $r$ times along the input dimension, giving $\widehat{\mathbf{W}}^L \in \mathbb{R}^{C_L \times rC_{L-1} \times K \times K}$. As for the intermediate layers, we scale $\mathbf{W}^l$ by $1/r$ and replicate the scaled weights $r$ times along both output and input channel dimensions, yielding $rC_l$ output channels and $rC_{l-1}$ input channels. To avoid the trivial solution caused by symmetrical/identical replications during training, we introduce a small noise $\epsilon$ into each replication. The theoretical proof of the equivalence of channel-level inverse reparameterization is provided in Appendix C.

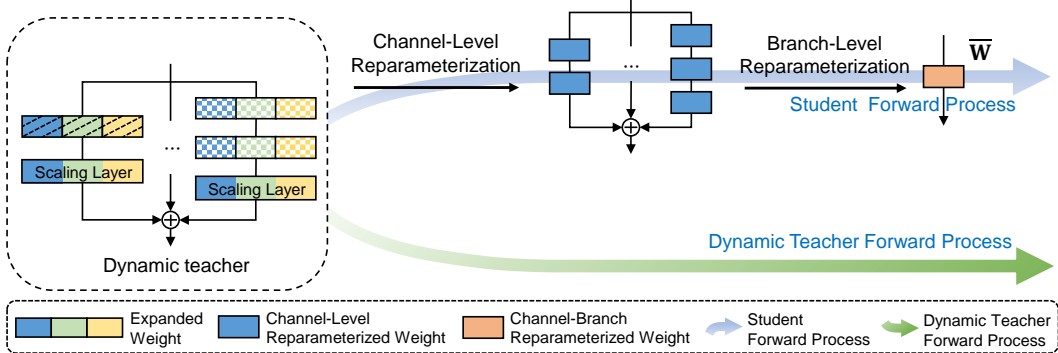

Figure 3: Illustration of the forward process for the student and dynamic teacher models. The dynamic teacher performs a direct forward pass, utilizing its increased capacity. The student model shares all parameters from the dynamic teacher and undergoes a two-step reparameterization process. First, channel-level reparameterization adjusts the expanded channels to match the original channel dimensions of the student model. Second, branch-level reparameterization merges the expanded multi-branch units into a single branch structure, thereby restoring the original topology of the student model while inheriting knowledge from the dynamic teacher.

### 3.2.2 BRANCH-LEVEL INVERSE REPARAMETERIZATION

Besides channel-level expansion, we also seek to expand the student model along the branch dimension. The branch-level inverse reparameterization aims to expand a single convolutional layer into a multi-branch structure with increased capacity, while preserving the identical input-output mapping. The key idea is to introduce additional branches with extra convolutions but zero-out them to keep the output of multi-branch architecture the same as the single-branch counterpart.

Specifically, considering a convolutional layer represented by the kernel $\mathbf{W} \in \mathbb{R}^{C_l \times C_{l-1} \times K \times K}$, where $\mathbf{X} \in \mathbb{R}^{C_{l-1} \times H \times W}$ and $\mathbf{Y} \in \mathbb{R}^{C_l \times H' \times W'}$ are the input and output tensors, respectively. Thus, the convolution operation can be represented by $\mathbf{Y} = \mathbf{W}\mathbf{X}$. As shown in Figure 1 (top right), we expand this single convolution into a multi-branch topology with $M$ branches. The first branch consists of a single convolutional layer, whose weights are initialized with the original convolution weights $\mathbf{W}$, followed by a learnable linear scaling layer $\mathbf{S}_1 \in \mathbb{R}^{C_l}$ initialized as a vector of ones. For the remaining $M - 1$ branches, each branch $m$ comprises a stack of convolutional layers, with their weights randomly initialized. All these branches are also followed by a learnable linear scaling layer $\mathbf{S}_m \in \mathbb{R}^{C_l}$, initialized with a vector of zeros. In this way, the output of the multi-branch block is equivalent to the original single-branch convolution. Let $f_m(\mathbf{X})$ denote the computation of the $m$-th branch. The expanded model along the branch dimension can be formulated by

$$\mathbf{Y} = \sum_{m=1}^{M} \mathbf{S}_m f_m(\mathbf{X}) = \mathbf{S}_1 f_1(\mathbf{X}) = \mathbf{W}\mathbf{X}. \tag{3}$$

### 3.3 PARAMETER SHARING WITH CHANNEL-BRANCH REPARAMETERIZATION

To further enhance knowledge transfer between the student $S$ and the dynamic teacher $T_d$, we propose a hard strategy for distillation that forces them to share parameters. In this way, the student is able to directly inherit well-trained parameters from a larger model, *i.e.*, dynamic teacher $T_d$. Based on the shared parameters, $S(x)$ and $T_d(x)$ have different predictions due to different forward propagation methods, as shown in Figure 3. Specifically, $T_d$ directly performs the forward pass, while the student $S$ conducts online reparameterization at both channel-level and branch-level.

### 3.3.1 CHANNEL-LEVEL REPARAMETERIZATION

As for channel-level reparameterization, given an expansion ratio $r$, we directly take the first $1/r$ kernels and scale them by $r$. Interestingly, this mapping is exactly the inverse transformation of how to conduct channel-level inverse parameterization (previously discussed in Section 3.2.1). Formally, let $\mathbf{W}_m^l \in \mathbb{R}^{rC_m^l \times rC_m^{l-1} \times K \times K}$ denote the weight of the $l$-th convolutional layer in the $m$-th branch of an expanded layer, and $rC_m^l$ and $rC_m^{l-1}$ are the numbers of output and input channels, respectively. During training, our channel-level reparameterization strategy explicitly extracts a subset of the dynamic teacher's expanded parameters to construct the student model's weights, enabling us to

obtain a promising student model after training. Specifically, we extract a channel-wise slice from $\mathbf{W}_m^l$ and apply a scaling operation:

$$\bar{\mathbf{W}}_m^l = r\mathbf{W}_m^l[: C_m^l, : C_m^{l-1}, :, :], \qquad (4)$$

where $\bar{\mathbf{W}}_m^l \in \mathbb{R}^{C_m^l \times C_m^{l-1} \times K \times K}$ denotes the reparameterized weight for the corresponding layer in the student model. The slicing operation $[: C_m^l, : C_m^{l-1}, :, :]$ extracts the first $C_m^l$ output channels and $C_m^{l-1}$ input channels from $\mathbf{W}_m^l$. The scaling ratio $r$ ensures the extracted parameters are appropriately scaled, aligning with the IR process applied during the construction of the dynamic teacher. For the first and last layers, the channel extraction is performed only over the output or input channel dimensions, respectively. Specifically, for the first layer, $\bar{\mathbf{W}}_m^1 = r\mathbf{W}_m^1[: C_m^1, :, :, :]$, while for the last layer, $\bar{\mathbf{W}}_m^L = \mathbf{W}_m^L[:, : C_m^{L-1}, :, :]$, as the last layer's weights are not scaled during IR process. Moreover, we found that parameter sharing with Batch Normalization (Ioffe & Szegedy, 2015) layers requires special treatment, as detailed in Appendix D.

### 3.3.2 Branch-Level Reparameterization

After channel-level reparameterization, we merge the expanded multi-branch units of the dynamic teacher model into the student's single-branch structure via branch-level reparameterization. Following Hu et al. (2022), for the $m$-th branch that contains $L_m$ convolutional layers $\mathbf{W}_m^1, \mathbf{W}_m^2, ..., \mathbf{W}_m^{L_m}$, we first obtain $\bar{\mathbf{W}}_m^1, \bar{\mathbf{W}}_m^2, ..., \bar{\mathbf{W}}_m^{L_m}$ through channel-level reparameterization. These are then merged into a single weight $\overline{\mathbf{W}}_m$ via the standard reparameterization, which is mathematically equivalent, as proven in (Ding et al., 2021b), by conducting sequential convolution operations: $\overline{\mathbf{W}}_m = \bar{\mathbf{W}}_m^1 \bar{\mathbf{W}}_m^2 ... \bar{\mathbf{W}}_m^{L_m}$. Performing this for all $M$ branches yields $\overline{\mathbf{W}}_1, \overline{\mathbf{W}}_2, ..., \overline{\mathbf{W}}_M$. After that, we sum up all of them to get the final new weight $\overline{\mathbf{W}} = \sum_{m=1}^M \overline{\mathbf{W}}_m$ for this layer in the student model. Through the branch reparameterization strategy, the student model effectively inherits well-trained parameters from the dynamic teacher model.

### 3.4 Effect of the Proposed Designs in Controlling Performance Gap

We highlight that the proposed designs are necessary to control the performance gap between student and teacher. Indeed, there exists a straightforward approach that uses a randomly initialized dynamic teacher without either IR or parameter sharing. In fact, it works well for training models from scratch, but it fails in fine-tuning scenarios since the randomly initialized teacher can easily destroy the distillation of a pre-trained student model. This motivates our design of IR, which initializes the dynamic teacher to start with the same accuracy as the student, making our method effective for both training from scratch and fine-tuning settings. On the other hand, the parameter sharing mechanism is also important since it acts as a hard constraint to control the gap between student and dynamic teacher within a suitable range. In this way, it is hard for the dynamic teacher to become significantly better than the student since they share/optimize the same set of parameters. Interestingly, parameter sharing also benefits the training of the student since the gradients come from optimizing both the dynamic teacher and the student. In fact, this has been empirically verified. Our ablation study (Table 4) demonstrates that parameter sharing contributes an additional 0.35% accuracy improvement. Together with IR, our GPD is able to effectively maintain appropriate performance gaps throughout the entire training process.

## 4 Experiments

### 4.1 Distillation with a Static Teacher

We closely follow the settings of Zhao et al. (2022a), Chen et al. (2021b) and put details in Appendix. Table 1 shows the consistent superiority of our GPD across diverse architecture when training from scratch with a static teacher. Notably, GPD not only yields substantial accuracy improvements when combined with existing KD methods like ReviewKD and DKD, but also outperforms the latest state-of-the-art KD approaches. Specifically, in the ResNet34 → ResNet18 setting, GPD boosts ReviewKD from 71.61% to 72.50%, and improves DKD by 1.01%, reaching 72.71%. These results outperform the most recent methods. In ResNet50 → MobileNet, GPD enhances ReviewKD by 0.65% and DKD by 1.58%. Similar gains are also observed in the transformer-based setting RVT-S → RVT-Ti. These significant accuracy improvements across diverse architectures highlight the

Table 1: Comparison of the performance of various distillation methods across different architectures. "-" denotes the result that is not reported. A → B indicates a teacher model A distilling knowledge to a student model B. GPD consistently enhances the performance of standard distillation methods across diverse architectures.

| Model | Teacher → Student | | |
|---|---|---|---|
| | ResNet34 → ResNet18 | ResNet50 → MobileNet | RVT-S → RVT-Ti |
| Teacher | 73.31 | 76.16 | 81.69 |
| Student | 69.75 | 68.87 | 78.45 |
| KD (Hinton et al., 2015) | 70.66 | 68.58 | - |
| AT (Zagoruyko & Komodakis, 2017a) | 70.69 | 69.56 | - |
| OFD (Heo et al., 2019a) | 70.81 | 71.25 | - |
| CRD (Tian et al., 2020) | 71.17 | 71.37 | - |
| RKD (Park et al., 2019) | 70.40 | 68.5 | - |
| WSLD (Zhou et al., 2021) | 72.04 | 71.52 | - |
| SRRL (Yang et al., 2021) | 71.73 | 72.49 | - |
| SimKD (Chen et al., 2022) | 71.59 | 72.25 | - |
| DIST (Huang et al., 2022) | 72.07 | 73.24 | - |
| NKD (Yang et al., 2023) | 71.96 | 72.58 | - |
| CAT-KD (Guo et al., 2023) | 71.26 | 72.24 | - |
| KD+CTKD (Li et al., 2023) | 71.38 | 71.16 | - |
| MLKD (Jin et al., 2023) | 71.90 | 73.01 | - |
| KD+CTKD+LS (Sun et al., 2024) | 71.81 | 72.92 | - |
| DKD+LSKD (Sun et al., 2024) | 71.88 | 72.85 | - |
| MLKD+LSKD (Sun et al., 2024) | 72.08 | 73.22 | - |
| CKD (Zhu et al., 2024b) | 72.24 | 72.97 | - |
| ReviewKD (Chen et al., 2021b) | 71.61 | 72.56 | 78.92 |
| ReviewKD + GPD | **72.50 (+0.89)** | **73.21 (+0.65)** | **80.01 (+1.09)** |
| DKD (Zhao et al., 2022a) | 71.70 | 72.05 | 79.12 |
| DKD + GPD | **72.71 (+1.01)** | **73.63 (+1.58)** | **80.14 (+1.02)** |

Table 2: Comparison of training from scratch without a static teacher model. GPD* denotes our method using only the dynamic teacher for distillation. As a standalone method, our GPD* consistently improves student model performance across diverse architectures.

| Model | ResNet18 | MobileNet | RVT-Ti |
|---|---|---|---|
| Baseline | 70.07 | 71.68 | 78.45 |
| GPD* | **71.87 (+1.80)** | **73.07 (+1.39)** | **79.85 (+1.40)** |

Table 3: Performance comparison of fine-tuning. GPD* denotes distillation using only the dynamic teacher model without the static teacher. Our method consistently outperforms the fine-tuning baseline with longer training across various architectures.

| Model | ResNet18 | MobileNet | RVT-Ti |
|---|---|---|---|
| Pre-trained Model | 69.75 | 68.87 | 78.45 |
| Longer Training | 70.23 | 69.01 | 78.61 |
| GPD* | **71.12 (+0.89)** | **69.47 (+0.46)** | **78.84 (+0.23)** |

effectiveness of our proposed method. By introducing a dynamic teacher model to mitigate the substantial gap between the student and a powerful static teacher, our GPD enables the student to more effectively absorb knowledge from the teacher. This leads to substantial performance improvements when combined with existing KD methods, surpassing even the most recent distillation techniques.

## 4.2 TRAIN FROM SCRATCH

In this experiment, we train ResNet18 and MobileNet for 100 epochs, and RVT-Ti for 300 epochs. Table 2 illustrates the effectiveness of our GPD* method, which utilizes only the dynamic teacher for distillation, without a static teacher. Across various backbone architectures, GPD* consistently improves upon baseline models trained without knowledge distillation. For example, the ResNet-18 model achieves a significant accuracy boost from 70.07% to 71.87% with GPD*, indicating a notable 1.80% improvement. Similarly, with GPD*, the MobileNet achieves a noteworthy improvement from 71.68% to 73.07%. These results highlight the versatility and effectiveness of our proposed method, which not only enhances the performance of existing KD methods when a strong static teacher is available but also serves as an effective stand-alone training strategy in scenarios where pre-trained teacher models are unavailable.

## 4.3 MODEL FINE-TUNING

In this experiment, we fine-tune the pre-trained models for 50 epochs and set the initial learning rate to $0.1\times$ w.r.t. its base/standard value. As for our GPD*, the dynamic teacher is constructed via Inverse Reparameterization based on the student model itself. This process ensures that the dynamic teacher initially exhibits the same accuracy as the student. In Table 3, compared to the accuracy of

pre-trained model, GPD* consistently achieves performance improvements across different architectures. Moreover, GPD* outperforms the longer training approach by up to 0.89% for ResNet18, and similar performance gains are observed for MobileNet and RVT-Ti models as well.

## 5 FURTHER DISCUSSIONS

**Gap preserving and parameter sharing.** We investigate the contribution of the proposed accuracy gap preserving mechanism and the parameter sharing strategy in this part. The ablation study in Table 4 validates the critical role of these components in boosting performance. Individually, the gap preservation mechanism yields a 0.66% accuracy gain, demonstrating its effectiveness in guiding the student model's learning trajectory. When combined with parameter sharing, the synergistic effect leads to a substantial 1.01% improvement over the baseline, underscoring the significance of these key innovations in facilitating knowledge transfer within the GPD framework.

**Channel-branch reparameterization.** Table 5 provides insights into the impact of different inverse reparameterization strategies on the performance of our GPD. Both channel-level and branch-level reparameterization techniques individually contribute to performance improvements over the baseline DKD method, achieving accuracy gains of 0.86% and 0.61%, respectively. However, their combined application yields the highest performance boost, with a remarkable 1.01% accuracy gain. Utilizing both techniques enables the dynamic teacher to effectively guide the student model, maximizing knowledge transfer within the GPD.

Table 4: Ablation studies on preserving the accuracy gap and parameter sharing. We take DKD as the baseline method to distill knowledge from ResNet34 to ResNet18. Preserving the accuracy gap alone improves performance over the baseline, and combining it with parameter sharing yields further gains.

| Preserve Gap | Share Param | Acc. (%) |
|:---:|:---:|:---:|
| Baseline | | 71.70 |
| ✓ | | 72.36 (+0.66) |
| ✓ | ✓ | **72.71 (+1.01)** |

Table 5: Impact of inverse reparameterization level on performance. We take DKD as the baseline method to distill knowledge from ResNet34 to ResNet18. The combined channel-level and branch-level strategy achieves the highest accuracy.

| Channel-level | Branch-level | Acc. (%) |
|:---:|:---:|:---:|
| Baseline | | 71.70 |
| ✓ | | 72.56 (+0.86) |
| | ✓ | 72.31 (+0.61) |
| ✓ | ✓ | **72.71 (+1.01)** |

**Branch and channel expansion ratio.** Figure 4 (left) shows that increasing the number of branches from 1 to 6 gradually improves accuracy, peaking at $M = 6$. However, further increasing to 12 branches leads to reduced performance gains, indicating that too many branches may be hard to train and do not necessarily improve performance. Figure 4 (right) illustrates the impact of the channel expansion ratio $r$. Both $r = 2$ and $r = 3$ significantly outperform the DKD baseline, but improvement dramatically drops at $r = 4$. The main reason is that the performance gap would be very large again when we consider a large dynamic teacher. We highlight that our GPD method is highly practical and efficient, since the default setting with $M = 6$ branches and $r = 2$ channel expansion ratio generalizes well to all the considered scenarios. In practice, we recommend using $M = 6$ and $r = 2$ and this setting generalizes well to diverse scenarios. As demonstrated in Section 4, these configurations consistently yield significant improvements across various architectures and distillation scenarios. Based on the above, building the dynamic teacher with $M = 6$ and $r = 2$ can be used as a good gap. More critically, GPD consistently enhances baseline performance across different branch and channel expansion ratios, demonstrating its robustness to parameter choices. Thus, it is unnecessary to carefully tune these hyper-parameters due to the high robustness.

**Effect of teacher-student gap size.** As shown in Table 6, when using increasingly larger teacher models to distill ResNet18, performance initially improves and then deteriorates. Specifically, knowledge transfer is most effective when we choose ResNet101 as the teacher, with performance gains of up to 1.16%. However, as the gap further increases with the models ResNet152 and ConvNeXt_Base, the distillation effectiveness significantly drops. Similarly, Table 9 shows a similar phenomenon in the transformer architecture family when using larger teachers (RVT-S to ViT-L) to distill RVT-Ti. These observations across both architecture families justify that overly large performance gaps can indeed hinder the distillation performance. We highlight that this phenomenon

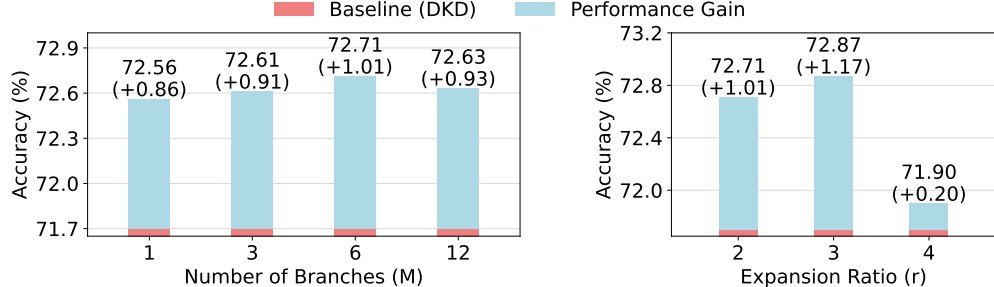

Figure 4: Impact of branch expansion number and channel expansion ratio on model accuracy. Performance gains are shown above the baseline (DKD). Left: Increasing $M$ to 6 yields significant improvements, with diminishing returns beyond that. Right: Channel expansion ratios of 2 and 3 show substantial gains, while a ratio of 4 leads to degradation.

Table 6: Impact of different teacher-student gap sizes on ResNet18 distillation performance. GPD consistently improves performance across varying teacher model sizes.

| Method | ResNet34 | ResNet101 | ResNet152 | ConvNeXt_Base | ViT-L |
|---|---|---|---|---|---|
| Teacher | 73.31 | 77.37 | 78.31 | 84.06 | 85.14 |
| Student | 69.75 | 69.75 | 69.75 | 69.75 | 69.75 |
| DKD | 71.70 | 71.74 | 71.61 | 71.49 | 71.43 |
| DKD + GPD | **72.71 (+1.01)** | **72.90 (+1.16)** | **72.81 (+1.20)** | **72.78 (+1.29)** | **72.71 (+1.28)** |

Table 7: Accuracy drop comparison when distilling ResNet18 from different teacher models (ResNet34 vs. ViT-L).

| Method | ResNet34 | ViT-L (Acc. Drop) |
|---|---|---|
| Teacher | 73.31 | 85.14 |
| Student | 69.75 | 69.75 |
| DKD | 71.70 | -0.27 (71.43) |
| DKD + GPD | 72.71 | **-0.00** (72.71) |

aligns well with both our empirical results and has been also observed in a lot of works (Son et al., 2021; Yang et al., 2019b; Mirzadeh et al., 2020). For a more intuitive illustration of the performance impact of the teacher-student gap, please refer to the Appendix G. On the other hand, we highlight that our GPD effectively mitigates the issue of large teacher-student performance gaps across various teacher model sizes. As shown in Table 7, when using ViT-L instead of ResNet34 as the teacher, DKD suffers a notable 0.27% accuracy drop (71.70→71.43), while GPD maintains the same accuracy (72.71%). Similarly, as presented in Table 8, when switching from ResNet-101 to ViT-L, DKD incurs a larger 0.31% accuracy decline (71.74→71.43), whereas GPD shows only a minor 0.19% drop (72.90→72.71). These results demonstrate GPD's effectiveness in handling large teacher-student gaps, ensuring stable student performance even when the teacher model is significantly larger.

Table 8: Accuracy drop comparison when distilling ResNet18 from different teacher models (ResNet101 vs. ViT-L).

| Method | ResNet101 | ViT-L (Acc. Drop) |
|---|---|---|
| Teacher | 77.37 | 85.14 |
| Student | 69.75 | 69.75 |
| DKD | 71.74 | -0.31 (71.43) |
| DKD + GPD | 72.90 | **-0.19** (72.71) |

Table 9: Performance of distilling RVT-Ti using transformer teachers of varying sizes. While overly large performance gaps can hinder distillation performance, GPD consistently improves performance across different teacher model capacities.

| Model | RVT-S | RVT-Base | ViT-L |
|---|---|---|---|
| Teacher | 81.69 | 82.51 | 85.15 |
| Student | 78.45 | 78.45 | 78.45 |
| DKD | 79.12 | 79.17 | 78.97 |
| DKD + GPD | **80.14 (+1.02)** | **80.27 (+1.10)** | **80.16 (+1.19)** |

## 6 CONCLUSION

In this paper, we propose Gap Preserving Distillation (GPD), a novel approach to bridging the accuracy gap between teacher and student models for more effective knowledge transfer. Our key contribution is the introduction of a dynamic teacher model that preserves an appropriate accuracy lead over the student during training. We propose Inverse Reparameterization to losslessly expand the student model along the channel and branch dimensions, constructing the dynamic teacher with increased capacity. Furthermore, we devise a parameter sharing strategy based on Channel-Branch Reparameterization, enabling the student to inherit parameters from the expanded dynamic teacher. This reduces computational costs while allowing the student to benefit from the teacher's enriched knowledge representations. The improved efficiency and performance of compact models facilitated by our approach could enable the deployment of deep learning solutions in resource-constrained environments, thereby promoting wider accessibility to AI technologies. Comprehensive experiments on the ImageNet dataset validate the effectiveness of GPD in boosting the performance of standard knowledge distillation methods across various backbone architectures.

## 7 ACKNOWLEDGEMENTS

This work is supported by the National Natural Science Foundation of China (Grant No. 62376099) and Natural Science Foundation of Guangdong Province, China (Grant No. 2024A1515010989).

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

APPENDIX

## A  OVERVIEW AND OUTLINE

In this paper, we propose Gap Preserving Distillation (GPD) to bridge the performance gap between large teacher and compact student models. GPD trains a dynamic teacher alongside the student, maintaining a reasonable gap throughout. We utilize parameter sharing and establish mappings via *Inverse Reparameterization (IR)* and *Channel-Branch Reparameterization (CBR)*. The supplementary material provides detailed theoretical analyses and additional experimental information to support the main paper, organized as follows:

- In Section B, we present a theoretical analysis of the teacher-student performance gap based on information bottleneck theory.

- In Section C, we provide mathematical proofs for Channel-Level Inverse Reparameterization in both CNNs and Transformer architectures, demonstrating the equivalence between the expanded model and the original model through detailed derivations.

- In Section D, we discuss our strategy for parameter sharing in batch normalization, illustrating the importance of maintaining independent running statistics for student and dynamic teacher models to ensure stable training.

- In Section E, we detail our experimental settings, covering various training scenarios including distillation with static teachers, training from scratch and fine-tuning.

- In Section F, we provide additional comparisons with teacher assistant methods and self-distillation approaches, demonstrating GPD's superior performance.

- In Section G, we analyze how the teacher-student capacity gap affects model performance, showing GPD's effectiveness in mitigating performance degradation with large teachers.

- In Section H, We analyze the evolution of performance gap during training, providing insights into GPD's effectiveness.

- In Section I, we evaluate the computational overhead introduced by our proposed methods, providing comparative analyses that highlight the efficiency of our approach.

- In Section J, we discuss the key advantages of GPD over existing KD methods.

## B  THEORETICAL ANALYSIS OF THE TEACHER-STUDENT PERFORMANCE GAP IN KNOWLEDGE DISTILLATION

The theoretical explanation for the performance gap issue has been rigorously proved by Wang et al. (2022) (in Section 4) and we further derive the key theoretical analysis to justify our arguments. To be specific, based on the information bottleneck (IB) theory (Shwartz-Ziv & Tishby, 2017), the mutual information between two variables is defined as $I(\cdot, \cdot)$. In this sense, the training of deep neural networks aims to maximize the mutual information $I(Y; F)$ between learned features $F$ and ground truth $Y$ while minimizing the mutual information $I(X; F)$ with input data $X$. In the context of KD, the goal of effective knowledge transfer can be expressed as retaining high mutual information between the teacher and student networks (Ahn et al., 2019). Thus, the optimization goal for the student model in KD can be described as follows:

$$\min_s \{ I(X; F_s) - \beta I(Y; F_s) + \gamma |I(X; F_t) - I(X; F_s)| + \gamma |I(Y; F_t) - I(Y; F_s)| \}, \quad (5)$$

where $F_s$ and $F_t$ are the features extracted by the student and teacher networks, respectively. The terms $|I(X; F_t) - I(X; F_s)|$ and $|I(Y; F_t) - I(Y; F_s)|$ measure the information divergence between teacher and student networks across input and output representations. $\beta$ and $\gamma$ are positive hyper-parameters to control the importance of these terms.

We highlight that a highly accurate model is often **over-confident**, to have very large mutual information with the target $I(Y; F)$ but small mutual information with the input $I(X; F)$. Nevertheless, $I(X; F)$ can be viewed as a type of "dark knowledge" (Wang et al., 2022) that is also very important for effective KD. Here, we take the example mentioned in Wang et al. (2022) to illustrate this: "considering an image with a man driving a car, although it may be uniquely labeled into the "car"

category, it still contains features of the "people" category". Such weak but non-negligible features extracted from the input (measured by $I(X; F)$) are the most valuable knowledge for distilling student models. Intuitively, **a low-capacity student model should avoid becoming over-confident, which justifies why a highly accurate teacher model may hamper the distillation performance**. One possible solution is to maintain suitable mutual information with the input $I(X; F)$.

Theoretically, for a highly accurate teacher model, the mutual information with the target $I(Y; F_t)$ is very large, while the mutual information with the input $I(X; F_t)$ is relatively small. In this case, $I(X; F_t) - I(X; F_s)$ in the second term becomes negative and $I(X; F_t)$ is a constant that does not affect computing gradients for the student. Thus, we omit $I(X; F_t)$ and the second term becomes $\min_s \gamma |I(X; F_t) - I(X; F_s)| = \min_s \gamma(I(X; F_s) - I(X; F_t)) = \min_s \gamma I(X; F_s)$. As for the third term, $I(Y; F_t) - I(Y; F_s)$ is positive because $I(Y; F_t)$ is very large. Similarly, the term $I(Y; F_t)$ is a constant that will not affect computing gradients, we also omit it and the third term becomes $\min_s \gamma |I(Y; F_t) - I(Y; F_s)| = \min_s -\gamma I(Y; F_s)$. Based on the above, the objective can be reformulated as:

$$\min_s (1 + \gamma)I(X; F_s) - (\beta + \gamma)I(Y; F_s). \tag{6}$$

This formulation demonstrates that the teacher model tends to aggressively compress the input-related information (i.e. $(1 + \gamma)I(X; F_s)$), potentially causing the student model to lose valuable "dark knowledge" that is desired for effective KD. In contrast, a weak teacher model tends to have large mutual information with the target $I(Y; F_t)$ as well as large mutual information with the input $I(X; F_t)$. In this case, $I(X; F_t) - I(X; F_s)$ in the second term is often positive and this term becomes $\min_s \gamma |I(X; F_t) - I(X; F_s)| = \min_s \gamma(I(X; F_t) - I(X; F_s)) = \min_s -\gamma I(X; F_s)$. The overall objective becomes

$$\min_s (1 - \gamma)I(X; F_s) - (\beta + \gamma)I(Y; F_s). \tag{7}$$

Compared with the previous objective, the difference is that a highly accurate teacher accelerates the compression of the mutual information with the input $I(X; F_s)$, while a weak teacher alleviates this issue. Therefore, we can reasonably assume that: **more mutual information with the input data can be the main reason why a weak teacher model achieves better distillation performance than a highly accurate teacher model**.

This theoretical perspective motivates our proposed dynamic teacher model approach, which maintains a reasonable performance gap to optimize the knowledge transfer process.

## C PROOF OF CHANNEL-LEVEL INVERSE REPARAMETERIZATION

### C.1 CHANNEL-LEVEL INVERSE REPARAMETERIZATION FOR CNNS

To mathematically prove the equivalence of channel-level inverse reparameterization for CNNs, consider a model with three convolutions (see Figure 2 (b)), where the parameters are denoted as $W^1$, $W^2$, and $W^3$. Using $\otimes$ to represent the convolutional operation, the computation of this model becomes

$$Y = W^3 \otimes W^2 \otimes W^1 \otimes X. \tag{8}$$

Following the expansion rules in Figure 2 (a), when performing channel-level expansion with a ratio of 2, we expand the number of output channels in the first two convolutions $W^1$ and $W^2$, scaling them by 1/2. For the Last convolution $W^3$, we expand the number of input channels without any scaling. For simplicity, we omit the introduced noise $\epsilon$ introduced to the expanded weights. Thus, the outputs of these three convolutions can be derived as follows:

$$Y_1 = \begin{bmatrix} W^1/2 \\ W^1/2 \end{bmatrix} \otimes X \tag{9}$$

$$Y_2 = \begin{bmatrix} W^2/2 & W^2/2 \\ W^2/2 & W^2/2 \end{bmatrix} \otimes Y_1$$

$$= \begin{bmatrix} W^2/2 & W^2/2 \\ W^2/2 & W^2/2 \end{bmatrix} \otimes \begin{bmatrix} W^1/2 \\ W^1/2 \end{bmatrix} \otimes X \qquad (10)$$

$$= \begin{bmatrix} W^2 \otimes W^1/2 \\ W^2 \otimes W^1/2 \end{bmatrix} \otimes X$$

$$Y_3 = \begin{bmatrix} W^3 & W^3 \end{bmatrix} \otimes Y_2$$

$$= \begin{bmatrix} W^3 & W^3 \end{bmatrix} \otimes \begin{bmatrix} W^2 \otimes W^1/2 \\ W^2 \otimes W^1/2 \end{bmatrix} \otimes X \qquad (11)$$

$$= W^3 \otimes W^2 \otimes W^1/2 \otimes X + W^3 \otimes W^2 \otimes W^1/2 \otimes X$$

$$= W_3 \otimes W_2 \otimes W_1 \otimes X$$

Clearly, given the same input $X$, the expanded model has the same output as that of the original model, *i.e.*, $Y_3 = Y$.

## C.2 CHANNEL-LEVEL INVERSE REPARAMETERIZATION FOR TRANSFORMER ARCHITECTURES

Our inverse re-parameterization (IR) can naturally extend to transformer architectures, as their key components (Feed-Forward Networks and Multi-Head Attention) primarily consist of linear layers, which can be mathematically viewed as 1×1 convolutions. One key difference lies in the Multi-Head Attention (MHA) module. Note that our re-parameterization would increase the number of channels in $QKV$ linear projections. Nevertheless, the expanded channels of $QKV$ features/outputs are not directly used to compute attention. Instead, we use the expanded channels to expand the head dimension, i.e., increasing the number of attention heads while maintaining the same dimension per head as the student model. In this way, it can be guaranteed that the computation in each head in the dynamic teacher still remains as consistent as possible with the one in the student. As for the re-parameterization process, we follow the strategy depicted in Figure 3 and directly omit the extra/expanded attention heads to obtain the student model from the dynamic teacher.

We illustrate this extension by considering a two-head attention module with an expansion ratio of 2 (omitting the noise term $\epsilon$ for clarity). As shown in Figure 5:

**1) Student model**: For an input X with two attention heads, the computation is defined as:

$$\text{Head}_i = \text{Attention}(XW_i^Q, XW_i^K, XW_i^V)$$

$$= \text{Attention}(Q_i, K_i, V_i) \qquad (12)$$

$$= \text{softmax}\left(\frac{Q_i(K_i)^T}{\sqrt{d_k}}\right) V_i$$

The final output is then computed by concatenating the attention heads and applying a projection:

$$Y = \text{Concat}(\text{Head}_1, \text{Head}_2)W^O \qquad (13)$$

**2) Expanded Model**: Following our re-parameterization principle, we duplicate and scale the Query, Key, and Value projection matrices: $\begin{bmatrix} W_i^Q/2 & W_i^Q/2 \end{bmatrix}$, $\begin{bmatrix} W_i^K/2 & W_i^K/2 \end{bmatrix}$, $\begin{bmatrix} W_i^V/2 & W_i^V/2 \end{bmatrix}$. These duplicated weights form new attention heads, doubling the number of heads while maintaining the same dimension per head. Consequently, the resulting Query, Key, and Value are halved to those of the student model. To preserve mapping equivalence, the Query and Key are scaled by 2 before the dot product to ensure identical softmax output, while the Value

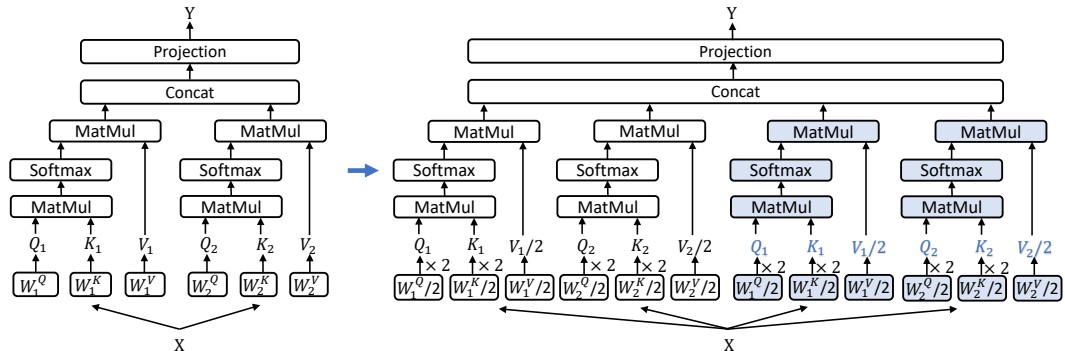

Figure 5: Illustration of channel-level inverse reparameterization in Multi-Head Attention (MHA). The student model (left) with 2 heads is expanded to 4 heads (right) by duplicating and scaling the Query, Key, and Value projection matrices, while preserving the original input-output mapping.

remains halved to maintain proper scaling of each head's output. This results in:

$$
\begin{aligned}
\text{Head}_i' &= \text{Attention}(X(W_i^Q/2)2, X(W_i^K/2)2, X(W_i^V/2)) \\[2mm]
&= \text{Attention}(Q_i, K_i, V_i/2) \\[2mm]
&= \text{softmax}\left(\frac{Q_i(K_i)^T}{\sqrt{d_k}}\right)V_i/2 \\[2mm]
&= \text{Head}_i/2
\end{aligned}
\tag{14}
$$

To demonstrate the equivalence between expanded model and student model, we consider the output projection as the last layer. Following the last-layer re-parameterization principle, the input dimension is expanded by duplicating $W^O$ without scaling (as detailed in Section 3.3.1), resulting in $\begin{bmatrix} W^O \\ W^O \end{bmatrix}$. This leads to the following equivalent final output:

$$
\begin{aligned}
Y' &= \text{Concat}(\text{Head}_1', \text{Head}_2', \text{Head}_1', \text{Head}_2')\begin{bmatrix} W^O \\ W^O \end{bmatrix} \\[2mm]
&= \text{Concat}(\text{Head}_1/2, \text{Head}_2/2, \text{Head}_1/2, \text{Head}_2/2)\begin{bmatrix} W^O \\ W^O \end{bmatrix} \\[2mm]
&= [\text{Concat}(\text{Head}_1, \text{Head}_2)/2 \quad \text{Concat}(\text{Head}_1, \text{Head}_2)/2]\begin{bmatrix} W^O \\ W^O \end{bmatrix} \\[2mm]
&= Y
\end{aligned}
\tag{15}
$$

## D  PARAMETER SHARING FOR BATCH NORMALIZATION

Batch normalization plays a crucial role in alleviating the vanishing gradient problem during the training of deep neural networks, ensuring stable convergence. However, since the student and dynamic teacher models may exhibit different data distributions during the training process, directly sharing the running statistics (*i.e.*, running mean and running variance) of the batch normalization layers would lead to mutual interference between the two models, potentially causing training instability. To avoid this issue, we propose maintaining independent running means and variances ($\tilde{\boldsymbol{\mu}}_s$, $\tilde{\boldsymbol{\sigma}}_s^2$) and ($\tilde{\boldsymbol{\mu}}_t$, $\tilde{\boldsymbol{\sigma}}_t^2$) for the student and dynamic teacher models, rather than sharing the statistics.

As illustrated in Figure 6, each model independently calculates its batch mean and batch variance from the input data, denoted as ($\boldsymbol{\mu}_s$, $\boldsymbol{\sigma}_s^2$) for the student model and ($\boldsymbol{\mu}_t$, $\boldsymbol{\sigma}_t^2$) for the dynamic teacher model. Subsequently, normalization is performed using these statistics, and the respective running statistics ($\tilde{\boldsymbol{\mu}}_s$, $\tilde{\boldsymbol{\sigma}}_s^2$) and ($\tilde{\boldsymbol{\mu}}_t$, $\tilde{\boldsymbol{\sigma}}_t^2$) are updated accordingly. The normalized outputs are then scaled

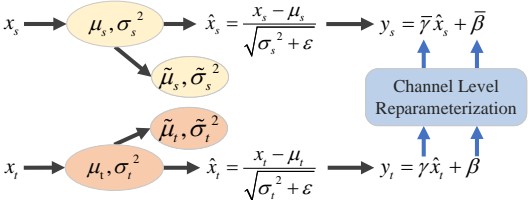

Figure 6: Illustration of parameter sharing for batch normalization in the student and dynamic teacher models. We maintain a separate set of running statistics for each model due to the distribution difference.

and shifted using the learnable parameters $\gamma$ and $\beta$, which are shared across the two models through the channel ensemble strategy. This design effectively eliminates potential instabilities arising from sharing statistics, thereby guaranteeing the robustness of the entire training process.

# E EXPERIMENTAL SETTINGS

## E.1 DISTILLATION WITH A STATIC TEACHER

In this experiment, we adopt the standard data pre-processing pipeline, including random cropping, resizing to 224×224, random horizontal flipping, and normalization. By default, we employ the SGD optimizer with an initial learning rate of 0.1 and a momentum of 0.9. For convolutional neural networks, the batch size is set to 256 on 4 Nvidia Tesla V100 GPUs, while for vision transformers (ViTs), the batch size is set to 256 on 8 Nvidia Tesla V100 GPUs. The models are trained for 100 epochs with a learning rate decay factor of 0.1 applied every 30 epochs. The weight decay is set to 1e-4, and the weight for the KD loss between the student and the dynamic teacher is set to 3.0, while the weights for the other KD losses and the cross-entropy loss are both set to 1.0. For convolutional neural networks, we strictly follow the settings from Zhao et al. (2022a); Chen et al. (2021b). For the same architecture family, the teacher model is ResNet-34, and the student model is ResNet-18. For different architecture families, the teacher model is ResNet-50, and the student model is MobileNet-V1. Additionally, we explore the vision transformer architecture RVT Mao et al. (2022), employing RVT-S as the teacher model and RVT-Ti as the student model. Notably, in our RVT experiments, we did not apply branch reparameterization. We found that channel reparameterization alone performed better than when branch reparameterization was included.

## E.2 TRAIN FROM SCRATCH

To further evaluate the efficacy of our approach, we conducted experiments without the reliance on a pre-trained static teacher model. Instead, we construct the dynamic teacher model via Inverse Reparameterization of the student model, as described in Sec. 3.2. Both the dynamic teacher and the student model are trained simultaneously from random initialization. During training, we leverage the additional loss terms introduced by our GPD, given by Eq. 2, to facilitate knowledge transfer from the dynamic teacher to the student model. We adopt the same data preprocessing and optimization strategies as described in Sec. 4.1.

## E.3 MODEL FINE-TUNING

In this part, we begin with pre-trained student models and aim to further improve their performance through our proposed approach. The pre-trained models are fine-tuned for 50 epochs, with the initial learning rate set to 0.1x the initial learning rate used during the pretraining stage. For the standard fine-tuning baseline, we fine-tune the pre-trained models with cross entropy loss. Regarding our proposed GPD* method, we leverage the dynamic teacher model for distillation-based fine-tuning. Benefiting from Inverse Reparameterization, the dynamic teacher model initially exhibits the same accuracy as the student model. During the fine-tuning process, the dynamic teacher maintains a slightly higher accuracy than the student due to its increased capacity. We used the same loss function as in the experiment of training from scratch.

## F   ADDITIONAL COMPARISONS WITH RELATED WORK

**Comparison with Teacher Assistant Methods.**   We compare with the state-of-the-art teacher-assistant-based KD methods. As shown in Table 10, GPD consistently achieves superior performance compared to these methods on ResNet18, demonstrating its effectiveness.

**Comparison with Self-Distillation Methods.**  To provide more comparisons when training without static teachers, we evaluate GPD* against state-of-the-art self-distillation methods. As shown in Table 11, GPD* consistently achieves superior performance compared to these methods on ResNet18, further demonstrating its effectiveness.

Table 10: Comparison with KD methods using teacher assistants on ImageNet.

| Method | ResNet18 |
| --- | --- |
| Student | 69.75 |
| TAKD (Mirzadeh et al., 2020) | 71.37 |
| DGKD (Son et al., 2021) | 71.73 |
| RKD (Gao et al., 2021) | 71.46 |
| TDS (Li et al., 2024a) | 72.29 |
| ESKD+AT (Cho & Hariharan, 2019) | 71.39 |
| TLLM (Zhu et al., 2022) | 72.6 |
| DKD + GPD | **72.71** |

Table 11: Comparison with Self-Distillation Methods on ImageNet.

| Method | ResNet18 |
| --- | --- |
| SSKD (Xu et al., 2020) | 71.62 |
| USKD (Yang et al., 2023) | 70.79 |
| FRSKD (Ji et al., 2021) | 70.17 |
| BYOT (Zhang et al., 2019) | 68.93 |
| ONE (Lan et al., 2018) | 70.55 |
| SD (Zhang et al., 2021) | 70.63 |
| RSD (Zheng et al., 2024) | 70.70 |
| GPD* | **71.87** |

## G   EFFECT OF TEACHER-STUDENT CAPACITY GAP ON MODEL PERFORMANCE

As visualized in Figure 7, increasing teacher capacity within an appropriate range can bring performance gains. However, DKD shows a clear declining trend as the teacher-student gap grows larger, suffering significant accuracy degradation with extremely large teachers like ViT-L. In contrast, GPD shows only minor performance decline and effectively alleviates the large-gap challenge, consistently outperforming DKD by more than 1% across all teacher models, even with the largest teacher ViT-L.

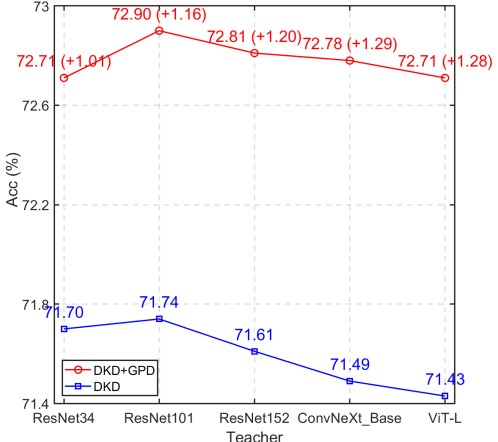

Figure 7:   Performance comparison with increasingly larger teacher models. DKD peaks with ResNet101 but suffers significant degradation with excessively large teachers, while GPD effectively mitigates performance degradation and maintains consistent improvements (+>1%).

## H   PERFORMANCE GAP EVOLUTION DURING DISTILLATION

The dynamic teacher often has higher accuracy during most of the training process except the very early stage. As shown in Figure 8 (distill ResNet18 from ResNet34), the dynamic teacher has lower/similar accuracy than the student in the first 3% epochs but outperforms it in the following training stages. Since our GPD maintains a reasonable performance gap in most of the training process, we are able to enhance the distillation performance by preserving a reasonable performance gap. More critically, we observe that this phenomenon exists in all the considered experiments.

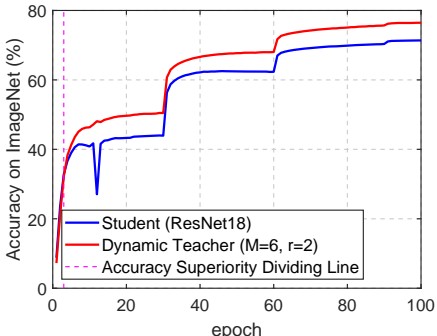

Figure 8: Convergence of student and dynamic teacher during ResNet18 distillation with GPD. The dynamic teacher shows lower/similar accuracy in the first 3% of epochs but outperforms the student in the following training stages.

## I   ANALYSIS OF COMPUTATIONAL OVERHEAD IN GPD

We highlight that the re-parameterization operation is very efficient and the time cost is negligible. To be specific, for a convolutional layer with 128 input channels and 256 output channels, it only takes 0.021 milliseconds, which is negligible compared to the overall computation cost. Our GPD slightly increases the training cost but comes with significant performance improvement. For example, from Table 12, GPD only introduces 33% extra overhead while yielding a large performance improvement of 1.58%. In fact, the additional training cost primarily arises from training the dynamic teacher model. Interestingly, the extra training overhead is marginal due to two reasons: 1) Since KD often relies on a large static teacher, conducting forward propagation through this model at each iteration already greatly increases the training cost; 2) Training both the student and dynamic teacher can increase the utilization ratio of GPU and benefits from its high parallelism in computation. Interestingly, we observe that merely training a small student model tends to come with a very low utilization ratio, e.g., often lower than 50%. Because our GPD consistently obtains significant performance improvement across diverse architectures and settings (see Table 1), we believe that it is a good trade-off between performance and training cost. From this point of view, our GPD could be a very strong baseline for distillation.

Table 12: Comparison of computation cost on the experiment of ResNet50 → MobileNet distillation. We measure the training time on 4 A100 GPUs with a batch size of 512 on ImageNet.

| Method | Acc. (%) | Training Time per Epoch (min) |
| --- | --- | --- |
| DKD | 72.05 | 12 |
| DKD + GPD | 73.63 (+1.58) | 16 (+33%) |

## J   ADVANTAGES OVER EXISTING METHODS

Our GPD has specific advantages and is essentially different from existing distillation methods (Wang et al., 2022; Mirzadeh et al., 2020; Cho & Hariharan, 2019; Yang et al., 2019a; Son et al., 2021; Zhao et al., 2022b) in several aspects. **First**, they maintain different performance gaps. Existing methods (Dong et al., 2024; Mirzadeh et al., 2020; Son et al., 2021) utilize fixed-accuracy pre-trained teachers assistant models, which may hinder knowledge transfer as the gap still be too large, particularly in early training stages when student accuracy is low. GPD initializes the dynamic teacher with the same accuracy as the student and trains both simultaneously, maintaining a small, dynamic gap throughout the process. **Second**, they construct the teacher assistants in different ways. Unlike methods (Mirzadeh et al., 2020; Son et al., 2021) that construct and train multiple intermediate-size teacher assistants separately, which can be computationally expensive, GPD builds a single dynamic teacher. We propose a novel IR technique for model expansion that maintains the same initial accuracy as the student. This approach is both computationally efficient and ensures a controlled start point for the distillation process. **Third**, they transfer knowledge in different ways. Existing methods (Mirzadeh et al., 2020; Son et al., 2021) follow the standard distillation paradigm by incorporating a KD loss for transferring knowledge. Besides this, our GPD enforces parameter sharing between the student and dynamic teacher, allowing direct inheritance of parameters. This process is facilitated by our CBR, enabling a more direct and effective knowledge transfer.

