# OpenReview forum: "Gap Preserving Distillation by Building Bidirectional Mappings with A Dynamic Teacher"
_ICLR.cc/2025/Conference — ICLR 2025 Poster_

### Official Review · Reviewer_FcyL · 2024-10-17

**Soundness:** 3
**Presentation:** 3
**Contribution:** 3
**Rating:** 6
**Confidence:** 4

**Summary:**

This paper aims to improve the effectiveness of knowledge distillation (KD) when using a teacher model with a large performance gap relative to the student model. The authors propose a method called Gap Preserving Distillation (GPD). By adopting a dynamic teacher model created through a re-parameterization operation on the student model as a proxy teacher, the performance gap is narrowed. Experimental results demonstrate the effectiveness of this approach.

**Strengths:**

- The problem of effectively performing KD with an extremely large teacher model is important and well-motivated.
- The paper is well-organized and easy to follow.

**Weaknesses:**

1. The abstract is too lengthy, as it seems to emphasize every aspect in detail. I suggest revising the abstract to improve clarity and conciseness.

2. The KD process involves training a dynamic teacher model alongside the student, which requires frequent re-parameterization. I am curious about the training cost compared to baseline methods.

3. In Table 6, it appears that the larger ViT-L teacher model leads to worse student performance than the smaller ResNet-101 teacher model (as with DKD). This suggests that the proposed method may still struggle with extremely large teacher models, despite it aims to overcome this issue.

4. The method's description primarily focuses on convolutional models. Although experimental results are provided for transformer models, it would be helpful to explain how re-parameterization is achieved for these models.

**Questions:**

Please refer to the weaknesses.

---

> ### Author Response · Authors · 2024-11-23
> **Authors' Response to Reviewer FcyL (Part 1)**
>
> We greatly appreciate the valuable feedback from the reviewer. We thank the reviewer for acknowledging the motivation behind our work, good organization, and clarity of our paper. We have carefully revised the paper according to your suggestions. Moreover, we highlight that our Inverse Reparameterization (IR) is essentially a new initialization method and generalizes well to diverse architectures. In fact, IR can be theoretically guaranteed and could be potentially useful in more tasks beyond distillation. We will release the code and all the models. Below, we provide detailed responses to the raised concerns.
>
> **Q1. The abstract is too lengthy, as it seems to emphasize every aspect in detail. I suggest revising the abstract to improve clarity and conciseness.**
>
> Thank you for your valuable suggestion. We have shortened the abstract by reducing the number of lines from 24 to 17. Please refer to the revised paper for the updated version.
>
> **Q2. The KD process involves training a dynamic teacher model alongside the student, which requires frequent re-parameterization. I am curious about the training cost compared to baseline methods**.
>
> - We highlight that the re-parameterization operation is very efficient and the time cost is negligible. To be specific, for a convolutional layer with 128 input channels and 256 output channels, it only takes 0.021 milliseconds, which is negligible compared to the overall computation cost.
>
> - Our GPD slightly increases the training cost but comes with significant performance improvement. For example, from Table 12 in the supplementary (also put it below for convenience), GPD only introduces 33\% extra overhead while yielding a large performance improvement of 1.58\%. In fact, the additional training cost primarily arises from training the dynamic teacher model. Interestingly, the extra training overhead is marginal due to two reasons: 1) Since KD often relies on a large static teacher, conducting forward propagation through this model at each iteration already greatly increases the training cost; 2) Training both the student and dynamic teacher can increase the utilization ratio of GPU and benefits from its high parallelism in computation. Interestingly, we observe that merely training a small student model tends to come with a very low utilization ratio, e.g., often lower than 50\%. Because our GPD consistently obtains significant performance improvement across diverse architectures and settings (see Table 1 in the paper), we believe that it is a good trade-off between performance and training cost. From this point of view, our GPD could be a very strong baseline for distillation. We have included the above in Appendix I.
>
> ***Table 12**: Comparison of computation cost on the experiment of ResNet50 → MobileNet distillation. We measure the training time on 4 A100 GPUs with a batch size of 512 on ImageNet.*
>
> |Method|Acc. (%)|Training Time per Epoch (min)|
> |:-:|:-:|:-:|
> |DKD|72.05|12|
> |DKD + GPD|73.63 (+1.58)|16 (+33%)|
>
> **Q3. In Table 6, it appears that the larger ViT-L teacher model leads to worse student performance than the smaller ResNet-101 teacher model (as with DKD). This suggests that the proposed method may still struggle with extremely large teacher models, despite it aims to overcome this issue.**
>
> Thanks for your valuable comment. We highlight that, although the issue of large teacher-student performance gaps still exists, GPD is able to effectively mitigate this issue across various teacher model sizes. As shown in Table G, when using ViT-L instead of ResNet34 as the teacher, DKD suffers a notable 0.27\% accuracy drop (71.70→71.43), while GPD maintains the same accuracy (72.71\%). Similarly, as presented in Table H, when switching from ResNet-101 to ViT-L, DKD incurs a larger 0.31\% accuracy drop (71.74→71.43), whereas GPD shows only a minor 0.19\% drop (72.90→72.71). These results demonstrate the effectiveness of GPD in handling large teacher-student gaps, ensuring stable student performance even when the teacher model is significantly larger. We have included these analyses in Table 7 and Table 8 of Section 5.
>
> ***Table G**: Accuracy drop comparison when distilling ResNet18 from different teacher models (ResNet34 vs. ViT-L).*
>
> |Method/Teacher|ResNet34|ViT-L (Acc. Drop)
> |:-:|:-:|:-:|
> |Teacher Acc.|73.31|85.14|
> |Student (ResNet18)|69.75|69.75|
> |DKD|71.70|-0.27 (71.43)|
> |DKD + GPD|72.71|**-0.00** (72.71)|
>
> ***Table H**: Accuracy drop comparison when distilling ResNet18 from different teacher models (ResNet101 vs. ViT-L).*
>
> |Method/Teacher|ResNet101|ViT-L (Acc. Drop)
> |:-:|:-:|:-:|
> |Teacher Acc.|77.37|85.14|
> |Student (ResNet18)|69.75|69.75|
> |DKD|71.74|-0.31 (71.43)|
> |DKD + GPD|72.90|**-0.19** (72.71)|

---

> ### Author Response · Authors · 2024-11-23
> **Authors' Response to Reviewer FcyL (Part 2)**
>
> **Q4. The method's description primarily focuses on convolutional models. Although experimental results are provided for transformer models, it would be helpful to explain how re-parameterization is achieved for these models.**
>
> Thank you for your constructive suggestion. Our inverse re-parameterization (IR) can naturally extend to transformer architectures, as their key components (Feed-Forward Networks and Multi-Head Attention) primarily consist of linear layers, which can be mathematically viewed as 1×1 convolutions. One key difference lies in the Multi-Head Attention (MHA) module. Note that our re-parameterization would increase the number of channels in $QKV$ linear projections. Nevertheless, the expanded channels of $QKV$ features/outputs are not directly used to compute attention. Instead, we use the expanded channels to expand the head dimension, i.e., increasing the number of attention heads while maintaining the same dimension per head as the student model. In this way, it can be guaranteed that the computation in each head in the dynamic teacher still remains as consistent as possible with the one in the student. As for the re-parameterization process, we follow the strategy depicted in Figure 3 and directly omit the extra/expanded attention heads to obtain the student model from the dynamic teacher.
>
> To illustrate this, we have added a detailed example in our revised paper showing how to expand a 2-head attention module with an expansion ratio of 2 (omitting the noise term ε for clarity). As shown in Figure 5 of the revised paper:
>
> **1) Student model**: For an input X with two attention heads, the computation is defined as:
> $$\text{Head}_i = \text{Attention}(XW^Q_i, XW^K_i, XW^V_i) = \text{Attention}(Q_i, K_i, V_i)=\text{softmax}\left(\frac{Q_i(K_i)^T}{\sqrt{d_k}}\right)V_i
> $$
> The final output is then computed by concatenating the attention heads and applying a linear projection:
> $$Y = \text{Concat}(\text{Head}_1, \text{Head}_2)W^O$$
> **2) Expanded Model**: Following our re-parameterization principle, we duplicate and scale the Query, Key, and Value projection matrices:
> $\begin{bmatrix}W^Q_i/2&W^Q_i/2\end{bmatrix}$, $\begin{bmatrix}W^K_i/2&W^K_i/2\end{bmatrix}$, $\begin{bmatrix}W^V_i/2&W^V_i/2\end{bmatrix}$.
> These duplicated weights form new attention heads, doubling the number of heads while maintaining the same dimension per head.  Consequently, the resulting Query, Key, and Value are halved to those of the student model. To preserve mapping equivalence, the **Query and Key are scaled by 2 before the dot product to ensure identical softmax output**, while the Value remains halved to maintain proper scaling of each head's output. This results in:
> $$\begin{aligned}
> \text{Head}'_i &= \text{Attention}(X(W^Q_i/2)*2, X(W^K_i/2)*2, X(W^V_i/2)) \\\\
>                 &= \text{Attention}(Q_i, K_i, V_i/2)\\\\
>                 &=\text{softmax}\left(\frac{Q_i(K_i)^T}{\sqrt{d_k}}\right)V_i/2\\\\
>                 &=\text{Head}_i/2
> \end{aligned}$$
> To demonstrate the equivalence between the expanded model and student model, we consider the output projection as the last layer. Following the last-layer re-parameterization principle, the input dimension is expanded by duplicating $W^O$ without scaling (as detailed in Section 3.2.1), resulting in $\begin{bmatrix}W^O\\\W^O\end{bmatrix}$. This leads to the following equivalent final output:
> $$\begin{aligned}
>     Y' &= \text{Concat}(\text{Head}'_1, \text{Head}'_2, \text{Head}'_1, \text{Head}'_2)\begin{bmatrix}W^O\\\W^O\end{bmatrix} \\\\
>     &= \text{Concat}(\text{Head}_1/2, \text{Head}_2/2, \text{Head}_1/2, \text{Head}_2/2)\begin{bmatrix}W^O\\\W^O\end{bmatrix} \\\\
>     &= \begin{bmatrix}\text{Concat}(\text{Head}_1, \text{Head}_2)/2&\text{Concat}(\text{Head}_1, \text{Head}_2)/2\end{bmatrix}
>     \begin{bmatrix}W^O\\\W^O\end{bmatrix} \\\\
>     &= Y
> \end{aligned}$$
>
> Due to our good generalization ability, we believe that IR is a **general initialization method and could be potentially useful in the community beyond distillation**. For example, if we want to get an extremely large pre-trained model (e.g., larger than 10B), it is possible to use our IR to directly initialize it from a well-trained small model that is much easier to train. Since our IR effectively preserves the accuracy/performance, the expanded model with a good initialization from IR could be beneficial to help converge faster for further finetuning. A typical use case could be finetuning large language models (LLMs) or extremely large diffusion models for AIGC. We will release the code to help reproduce our results.
>
>
> We have included detailed derivations and visual illustrations in Appendix C.2 of the revised paper to fully explain the implementation details of re-parameterization for transformer architectures.

---

> > ### Comment · Reviewer_FcyL · 2024-11-25
> >
> > Thank you for your response. Most of my concerns have been addressed. I have updated my score to 6.

---

> > > ### Author Response · Authors · 2024-11-25
> > > **Thanks for your feedback**
> > >
> > > Dear Reviewer FcyL,
> > >
> > > Thank you for your feedback! We greatly appreciate the constructive reviews and valuable suggestions to enhance our work.
> > >
> > > Best regards,
> > >
> > > Authors of #869

---

### Official Review · Reviewer_J6Y5 · 2024-11-01

**Soundness:** 3
**Presentation:** 3
**Contribution:** 3
**Rating:** 6
**Confidence:** 4

**Summary:**

The paper proposes Gap Preserving Distillation (GPD), a framework designed to address the capacity gap between teacher and student networks. GPD achieves this by training an additional dynamic teacher alongside the student. To further improve performance and efficiency, GPD enforces parameter sharing between the dynamic teacher and the student using Inverse Reparameterization and Channel-Branch Reparameterization techniques. Experiments on ImageNet-1K classification primarily compare GPD with baselines using static teachers, showing preliminary effectiveness.

**Strengths:**

1. The problem is well-motivated, particularly in lines 54-61, highlighting the importance of adaptively managing the performance gap.
2. The proposed approach is novel. It co-trains a dynamic teacher to alleviate the teacher-student capacity gap and applies inverse reparameterization and channel-branch reparameterization for efficient parameter sharing.
3. The paper is well-organized and easy to follow, with clear descriptions, intuitive figures, and rigorously detailed algorithms.

**Weaknesses:**

1. **Lack of comparisons with Knowledge Distillation (KD) methods using teacher assistants**. These methods are closely related to the proposed GPD. Without these comparisons, the effectiveness of GPD is unclear or less convincing. Suggested comparisons include, but are not limited to:
   - Mirzadeh, Seyed Iman, et al. "Improved knowledge distillation via teacher assistant." *AAAI 2020*
   - Son, Wonchul, et al. "Densely guided knowledge distillation using multiple teacher assistants." *CVPR 2021*
   - Ganta, Durga Prasad, Himel Das Gupta, and Victor S. Sheng. "Knowledge distillation via weighted ensemble of teaching assistants." *ICBK 2021*
   - Zhou, Yuhang, and Wei Ai. "Teaching-Assistant-in-the-Loop: Improving Knowledge Distillation from Imperfect Teacher Models in Low-Budget Scenarios." *arXiv:2406.05322* (2024)

2. **Inappropriate and unfair baselines in Section 4.2**. The paper compares GPD without a static teacher against networks trained without any KD techniques. Since GPD involves an additional dynamic teacher, it would be more appropriate to compare it with self-distillation methods.

3. **Practicality and efficiency concerns**. The size of the additional dynamic teacher must still be tuned by adjusting the branch and channel expansion ratios, making GPD less practical and efficient.

**Questions:**

1. The paper claims that the dynamic teacher shares exactly the same accuracy as the student with the reparameterization techniques. However, Figure 1 shows that the weights of the dynamic teacher are modified with added noise, which seems to contradict this claim.
2. The claim in lines 68-69, “Since DT is a larger model and often converges faster than the student,” appears counterintuitive. Is there any evidence to support this?
3. The network pairs in Table 6 do not convincingly demonstrate the effect of the teacher-student capacity gap. For instance, Table 6 shows that ViT-L performs worse than ResNet101 when distilling to a ResNet18. This may be due not to the larger capacity gap between ViT-L and ResNet18, but rather to differences in architecture. It would be more consistent to use CNN or ViT architectures for both teacher and student.

---

> ### Author Response · Authors · 2024-11-23
> **Authors' Response to Reviewer J6Y5  (Part 1)**
>
> We sincerely appreciate the valuable feedback from the reviewer. We thank the reviewer for recognizing that our work is well-motivated, acknowledging the novelty of our approach in managing performance gaps through dynamic teacher co-training, and commending the paper's clear organization and presentation. In our response and the revised manuscript, we provide extensive comparisons with teacher-assistant methods and self-distillation approaches; clarify the practicality of our method; and present additional evidence to demonstrate its effectiveness in alleviating the performance gap on top of consistent architecture families.
>
> **Q1**. **Lack of comparisons with Knowledge Distillation (KD) methods using teacher assistants.** These methods are closely related to the proposed GPD. Without these comparisons, the effectiveness of GPD is unclear or less convincing. Suggested comparisons include, but are not limited to:
>
> [D] Mirzadeh, Seyed Iman, et al. "Improved knowledge distillation via teacher assistant." AAAI 2020
>
> [E] Son, Wonchul, et al. "Densely guided knowledge distillation using multiple teacher assistants." CVPR 2021
>
> [F] Ganta, Durga Prasad, Himel Das Gupta, and Victor S. Sheng. "Knowledge distillation via weighted ensemble of teaching assistants." ICBK 2021
>
> [G] Zhou, Yuhang, and Wei Ai. "Teaching-Assistant-in-the-Loop: Improving Knowledge Distillation from Imperfect Teacher Models in Low-Budget Scenarios." arXiv:2406.05322 (2024)
>
> We appreciate your insightful suggestion and have included the comparisons with the state-of-the-art teacher-assistant-based KD methods, including TAKD [D], DGKD [E], RKD [H], TDS [I], ESKD+AT [J] and TLLM [K]. We highlight that we cannot directly compare with [F] and [G] due to the missing results and code. As for [F], this work does not release the code and only trains a two-layer network on small datasets (MNIST and CIFAR), without any results on ImageNet. Regarding [G], the authors do not release the code either. Since this work focuses on compressing large language models, we cannot directly compare with their reported results. As shown in Table C, our GPD consistently achieves superior performance, demonstrating its effectiveness. We have included these results in Table 10 in the supplementary.
>
> ***Table C**: Comparison of GPD with KD methods using teacher assistants. *
>
> |Method|ResNet18|
> |:-:|:-:|
> |Student|69.75|
> |TAKD [D]| 71.37 |
> |DGKD [E]| 71.73 |
> |RKD [H]|71.46|
> |TDS [I]|72.29|
> |ESKD+AT [J]|71.39|
> |TLLM [K]|72.6|
> |DKD + GPD|**72.71**|
>
> [H] Residual error based knowledge distillation. Neurocomputing 2021.
>
> [I] TAS: Distilling Arbitrary Teacher and Student via a Hybrid Assistant. arXiv 2024
>
> [J] On the Efficacy of Knowledge Distillation. ICCV 2019
>
> [K] Teach Less, Learn More: On the Undistillable Classes in Knowledge Distillation. NeurIPS 2022
>
>
> **Q2**. **Inappropriate and unfair baselines** in Section 4.2. The paper** compares GPD without a static teacher against networks trained without any KD techniques**. Since GPD involves an additional dynamic teacher, it would be more appropriate to compare it with **self-distillation methods**.
>
> Thanks for your valuable suggestion. We have included comparisons with state-of-the-art self-distillation methods, including SSKD [L], USKD [M], FRSKD [N], BYOT [O], ONE [P], SD [Q], and RSD [R]. As shown in Table D, GPD consistently achieves superior performance compared to these methods on ResNet18, further demonstrating its effectiveness. These results have been included in Table 11 in the supplementary to provide a more comprehensive evaluation.
>
> ***Table D**: Comparison of GPD\* with self-distillation methods.*
>
> |Model|ResNet18|
> |-|-|
> |Baseline|70.07|
> |SSKD [L]|71.62|
> |USKD [M]|70.79|
> |FRSKD [N]|70.17|
> |BYOT [O]|69.84|
> |ONE [P]|70.55|
> |SD [Q]|70.63|
> |RSD [R]|70.70|
> | **GPD\***|**71.87**|
>
> [L] Knowledge Distillation Meets Self-Supervision. ECCV 2022
>
> [M] From Knowledge Distillation to Self-Knowledge Distillation: A Unified Approach with Normalized Loss and Customized Soft Labels. ICCV 2023
>
> [N] Refine Myself by Teaching Myself: Feature Refinement via Self-Knowledge Distillation.CVPR 2021
>
> [O] Be Your Own Teacher: Improve the Performance of Convolutional Neural Networks via Self Distillation. ICCV 2019
>
> [P] Knowledge Distillation by On-the-Fly Native Ensemble. NeurIPS 2018
>
> [Q] Self-Distillation: Towards Efficient and Compact Neural Networks. TPAMI 2022
>
> [R] Restructuring the Teacher and Student in Self-Distillation. IEEE Transactions on Image Processing 2024

---

> ### Author Response · Authors · 2024-11-23
> **Authors' Response to Reviewer J6Y5  (Part 2)**
>
> **Q3. Practicality and efficiency concerns. The size of the additional dynamic teacher must still be tuned by adjusting the branch and channel expansion ratios, making GPD less practical and efficient**.
>
> We highlight that our GPD method is highly practical and efficient, since the default setting with M=6 branches and r=2 channel expansion ratio generalizes well to all the considered scenarios.
> As shown in Section 4, our GPD consistently obtains significant accuracy improvement across diverse architectures and distillation scenarios (e.g., CNNs, Transformers, with/without static teacher). More critically, as illustrated in Figure 4 of paper, GPD consistently enhances baseline performance across different branch and channel expansion ratios, demonstrating its robustness to parameter choices. Thus, it is unnecessary to carefully tune these hyper-parameters due to the high robustness.
>
>
> **Q4. The paper claims that the dynamic teacher shares exactly the same accuracy as the student with the reparameterization techniques. However, Figure 1 shows that the weights of the dynamic teacher are modified with added noise, which seems to contradict this claim.**
>
> Thanks for pointing it out. We have rephrased the argument "exactly the same accuracy" to "approximately the same accuracy" to make it clearer.
> In practice, the accuracy difference between the dynamic teacher and the original student model is only about 0.002%, which is often negligible. We highlight that the small Gaussian noise (mean 0, std 1e-6) added to the replicated weights merely seeks to break the weight symmetry issue, such that the copied weights would receive the same gradients. This minimal perturbation has no practical impact on model accuracy while enabling effective training of the expanded dynamic teacher.
>
>
> **Q5. The claim in lines 68-69, “Since DT is a larger model and often converges faster than the student,” appears counterintuitive. Is there any evidence to support this?**
>
> We apologize for the unclear argument in our original statement. We have rephrased it to "Since DT is a larger model and often has higher accuracy than the student". We highlight that it holds during most of the training process except the very early stage. As shown in Figure 8 in the supplementary (distill ResNet18 from ResNet34), the dynamic teacher has lower/similar accuracy than the student in the first 3\% epochs but outperforms it in the following training stages. Since our GPD maintains a reasonable performance gap in most of the training process, we are able to enhance the distillation performance by preserving a reasonable performance gap. More critically, we observe that this phenomenon exists in all the considered experiments. We have included these in Appendix H.
>
>
>
> **Q6. The network pairs in Table 6 do not convincingly demonstrate the effect of the teacher-student capacity gap. For instance, Table 6 shows that ViT-L performs worse than ResNet101 when distilling to a ResNet18. This may be due not to the larger capacity gap between ViT-L and ResNet18, but rather to differences in architecture. It would be more consistent to use CNN or ViT architectures for both teacher and student**.
>
> We appreciate this insightful suggestion. We highlight that the performance degradation is indeed caused by the large capacity gap rather than the difference in architectures. To rigorously validate this, we choose the student and teacher from the same architecture family and conduct experiments on CNNs and transformers, respectively. Within each family, we construct the teacher models with diverse capacities. As shown in Table E, when distilling ResNet18 with increasingly larger CNN teachers, the performance DKD greatly drops as the teacher model becomes larger than ResNet101. Similarly, Table F shows a similar phenomenon in the transformer architecture family when using larger teachers (RVT-S to ViT-L) to distill RVT-Ti. These observations across both architecture families justify that overly large performance gaps can indeed hinder the distillation performance. We highlight that this phenomenon aligns well with both our empirical results and has been also observed in a lot of works (Son et al., 2021; Yang et al., 2019b; Mirzadeh et al., 2020). We have included these results in the Table 6 and Table 9.
>
> ***Table E**: Comparison of distilling ResNet18 using increasingly larger teachers.*
>
> |Method|ResNet34|ResNet101|ResNet152|ConvNeXt_Base|
> |:-:|:-:|:-:|:-:|:-:|
> |Teacher|73.31|77.37|78.31|84.06|
> |Student|69.75|69.75|69.75|69.75|
> |DKD|71.70|71.74| 71.61|71.49 |
> |DKD + GPD| **72.71 (+1.01)**|**72.90 (+1.16)**|**72.81(+1.20)**|**72.78(+1.29)**|
>
>
> ***Table F**: Comparison of distilling RVT-Ti using increasingly larger teachers.*
>
> |Method|RVT-S|RVT-Base|ViT-L|
> |:-:|:-:|:-:|:-:|
> |Teacher|81.69|82.51|85.15|
> |Student|78.45|78.45|78.45|
> |DKD|79.12|79.17|78.97|
> |DKD + GPD| **80.14 (+1.02)**|**80.27 (+1.10)**|**80.16 (+1.19)**|

---

> > ### Author Response · Authors · 2024-11-25
> > **Kind Reminder to Reviewer J6Y5 for the Feedback on Our Rebuttal**
> >
> > Dear Reviewer J6Y5,
> >
> > Thanks for your thoughtful review and valuable comments. We notice that Reviewer FcyL has provided feedback to our rebuttal and we really appreciate such a fast response. During the discussion period, we also want to get some feedback from you.
> >
> > Actually, your comments are particularly insightful, and we believe they will help strengthen our work significantly. In our rebuttal, we have carefully addressed each of your concerns with detailed responses. Specifically, we have included **1)** comprehensive comparisons with teacher-assistant methods and self-distillation baselines, **2)** clarified the practicality of our approach, and **3)** presented additional evidence demonstrating its effectiveness in alleviating the performance gap across consistent architecture families. We would sincerely appreciate it if we could get some feedback from you regarding the above concerns. If you have any further questions or require additional clarifications, please do not hesitate to let us know.
> >
> > Thank you for your time and consideration.
> >
> > Best regards,
> >
> > Authors of #869

---

> > > ### Comment · Reviewer_J6Y5 · 2024-11-25
> > > **Thank you for the rebuttal**
> > >
> > > Thank you for the rebuttal, which well addresses most of my concerns and questions. I have updated my score from 5 to 6 accordingly.

---

> > > > ### Author Response · Authors · 2024-11-25
> > > > **Thanks for your feedback**
> > > >
> > > > Dear Reviewer J6Y5,
> > > >
> > > > Thank you for your feedback! We greatly appreciate the constructive reviews and valuable suggestions to enhance our work.
> > > >
> > > > Best regards,
> > > >
> > > > Authors of #869

---

### Official Review · Reviewer_49EM · 2024-11-06

**Soundness:** 3
**Presentation:** 3
**Contribution:** 2
**Rating:** 6
**Confidence:** 3

**Summary:**

This paper introduces Gap Preserving Distillation (GPD) to address the challenge where large performance gaps between teacher and student models hinder effective knowledge transfer. GPD introduces a dynamic teacher model that trains alongside the student, maintaining an optimal performance gap to improve transfer efficiency. The performance gains from GPD are primarily due to its use of Inverse Reparameterization (IR) and Channel-Branch Reparameterization (CBR), which create flexible connections that enable parameter sharing between the dynamic teacher and student. Experimental results show that models trained with GPD outperform those using traditional fixed teacher models and perform robustly even in scenarios without a pre-trained teacher, such as training from scratch and fine-tuning.

**Strengths:**

1. This paper identifies the performance gap between teacher and student models as a key challenge in knowledge distillation. By introducing a dynamic adjustment mechanism, the method effectively narrows this gap, facilitating more efficient knowledge transfer.

2. The proposed GPD demonstrates adaptability across various training settings, with significant improvements across multiple models and datasets. Extensive ablation studies further validate the method’s effectiveness.

**Weaknesses:**

1. Needs More Theoretical Explanation
This paper could benefit from a deeper explanation of why a large performance gap between the teacher and student models makes knowledge transfer less effective. Right now, the idea that a big gap harms learning is mentioned, but it would be stronger if some theoretical reasoning or tests showed how this happens. Adding experiments that test different gap sizes could also help show how different levels of teacher-student gaps impact the student model’s final performance.

2. Overengineering in the Dynamic Teacher Model
The necessity of the complex structure designed in this paper, including Inverse Reparameterization (IR) and parameter sharing, appears somewhat disconnected from the core problem. If the primary goal is to reduce the performance gap, it remains unclear whether the proposed architecture (specifically the IR and shared parameters) is required to achieve this. Making the connection between these design choices and the main goal a bit clearer could help the paper feel more cohesive.

**Questions:**

1. How can we better understand and measure the impact of the performance gap on knowledge transfer? The paper mentions that keeping a reasonable performance gap between teacher and student models helps with effective knowledge transfer. However, it doesn’t fully explain why a larger gap might hurt the process. How exactly does the size of this gap affect the student model’s final performance? As mentioned in the weaknesses section, adding some theory or controlled experiments here could help clarify this point and give a clearer idea of the best range for this gap.

2. As mentioned in the weaknesses, is the complex design of the dynamic teacher model really necessary? To tackle the performance gap, the paper introduces a dynamic teacher model with intricate structures like Inverse Parameterization (IR) and parameter sharing. But has the author considered more straightforward options to close the gap? If this complex setup is essential, could they explain more about how it directly addresses the problem?

---

> ### Author Response · Authors · 2024-11-23
> **Authors' Response to Reviewer 49EM (Part 1)**
>
> We sincerely thank the reviewer for the insightful feedback. Our response provides **1)** additional experiments, **2)** more discussions on performance gap measurement, and **3)** detailed justification of the necessity and effectiveness of our Inverse Reparameterization (IR) and parameter-sharing strategies in addressing the gap issue.
>
> **Q1. Theoretical explanation on why a large performance gap between teacher and student models hinders effective knowledge transfer.**
>
> Thanks for the valuable suggestion. The performance gap issue has been theoretically proved by [A] (in Section 4) and we further derive the key theoretical analysis to justify our arguments. To be specific, based on the information bottleneck (IB) theory [B], the mutual information between two variables is defined as $I(\cdot, \cdot)$. In this sense, the training of deep neural networks aims to maximize the mutual information $I(Y; F)$ between learned features $F$ and ground truth $Y$ while minimizing the mutual information $I(X; F)$ with input data $X$. In Knowledge Distillation (KD), the goal of effective knowledge transfer can be expressed as retaining high mutual information between the teacher and student networks [C]. Thus, the the student model's optimization goal in KD can be described as follows:
> $$
> \min_{s}\\{ I(X; F_s)-\beta I(Y; F_s)+\gamma|I(X; F_t) - I(X; F_s)|+\gamma|I(Y; F_t)-I(Y; F_s)|\\},
> $$
> where $F_s$ and $F_t$ are student and teacher features, respectively. The terms $|I(X; F_t) - I(X; F_s)|$ and $|I(Y; F_t)-I(Y; F_s)|$ measure the information divergence between teacher and student networks across input and output representations. $\beta$ and $\gamma$ are positive hyper-parameters to control the importance of these terms.
>
> We highlight that a highly accurate model is often **over-confident** to have very large mutual information with the target $I(Y; F)$ but small with the input $I(X; F)$. While $I(X; F)$ represents crucial "dark knowledge" [A] for KD, as illustrated by: *"considering an image with a man driving a car, although it may be uniquely labeled into the "car" category, it still contains features of the "people" category"*. Such weak but non-negligible features extracted from the input (measured by $I(X; F)$) are vital for student model distillation. Intuitively, **a low-capacity student model should avoid becoming over-confident, which justifies why a highly accurate teacher model may hamper the distillation performance**. One possible solution is to maintain suitable input mutual information $I(X; F)$. Interestingly, it can be theoretically proved that a weak teacher often comes with stronger mutual information with the input $I(X; F)$.
>
> Theoretically, a highly accurate teacher model has very large $I(Y; F)$ but relatively small $I(X; F)$. In this case, $I(X; F_t) - I(X; F_s)$ becomes negative and $I(X; F_t)$ is a constant that does not affect student gradients. Thus, we omit $I(X; F_t)$ and the second term becomes $\min_s \gamma|I(X; F_t) - I(X; F_s)| = \min_s \gamma(I(X; F_s) - I(X; F_t)) = \min_s \gamma I(X; F_s)$. For the third term, $I(Y; F_t)-I(Y; F_s)$ is positive because $I(Y; F_t)$ is very large. Similarly, the term $I(Y; F_t)$ is a constant that will not affect computing gradients, we also omit it and the third term becomes $\min_{s} \gamma|I(Y; F_t)-I(Y; F_s)| = \min_{s} - \gamma I(Y; F_s)$. Based on the above, the objective becomes:
> $$
> \min_{s}\\{(1 + \gamma) I(X; F_s) - (\beta + \gamma) I(Y; F_s) \\}.
> $$
> This shows the teacher model aggressively compresses the input-related information (i.e. $(1 + \gamma)I(X; F_s)$), potentially causing the student to lose valuable "dark knowledge" essential for effective KD. In contrast, a weak teacher tends to have large mutual information with the target $I(Y; F_t)$ as well as large mutual information with the input $I(X; F_t)$. In this case, $I(X; F_t) - I(X; F_s)$ in the second term is often positive, simplifying to $\min_s \gamma|I(X; F_t) - I(X; F_s)| = \min_s \gamma(I(X; F_t) - I(X; F_s)) = \min_s - \gamma I(X; F_s)$. The objective becomes
> $$
> \min_{s}\\{(1 - \gamma) I(X; F_s) - (\beta + \gamma) I(Y; F_s)\\}.
> $$
>
> Compared with the previous objective, the difference is that a highly accurate teacher accelerates the compression of the mutual information with the input $I(X; F_s)$, while a weak teacher alleviates this issue. Therefore, we can reasonably assume that: **more mutual information with the input data can be the main reason why a weak teacher model achieves better distillation performance than a highly accurate teacher model**.
>
> This theoretical perspective motivates our proposed GPD, which maintains a reasonable performance gap to optimize the knowledge transfer. we have included these in Appendix B.
>
> [A] Efficient Knowledge Distillation from Model Checkpoints. NeurIPS 2022
>
> [B] Opening the Black Box of Deep Neural Networks via Information. CoRR 2017
>
> [C] Variational information distillation for knowledge transfer. CVPR 2019

---

> ### Author Response · Authors · 2024-11-23
> **Authors' Response to Reviewer 49EM (Part 2)**
>
> **Q2. Adding experiments that test different gap sizes could also help, how different levels of teacher-student gaps impact the student model’s final performance. How exactly does the size of this gap affect the student model’s final performance?**
>
> Thanks for your valuable suggestions, we conduct experiments with different teacher-student combinations to analyze how the size of the teacher-student gap affects the student model’s final performance. As shown in Table A, when using increasingly larger teacher models to distill ResNet18, performance initially improves and then deteriorates. Specifically, knowledge transfer is most effective when we choose ResNet101 as the teacher, with performance gains of up to 1.16%. However, as the gap further increases with the models ResNet152 and ConvNeXt_Base, the distillation effectiveness significantly drops. This empirical evidence aligns with our theoretical analysis and demonstrates the importance of maintaining an appropriate gap during knowledge transfer.
>
> On the other hand, we highlight that our GPD is able to alleviate this issue even though we choose a teacher model with a very large performance gap.
> As shown in Table B, when using ViT-L, a substantially larger model with an 85.14% accuracy, DKD experiences a significant 0.27% accuracy drop (from 71.70% to 71.43%). In contrast, our GPD maintains the same accuracy as when using a relatively well-matched teacher ResNet34, effectively mitigating the challenges posed by large performance gaps.
>
> In the revised paper, we have included Figure 7 to visualize how the teacher with different capacities affects the performance of the student.
>
> ***Table A**: Impact of different teacher-student gap sizes on ResNet18 distillation performance.*
>
> |Method|ResNet34|ResNet101|ResNet152|ConvNeXt_Base|ViT-L|
> |:-:|:-:|:-:|:-:|:-:|:-:|
> |Teacher|73.31|77.37|78.31|84.06|85.14|
> |Student|69.75|69.75|69.75|69.75|69.75|
> |DKD|71.70|71.74| 71.61|71.49 |71.43|
> |DKD + GPD| **72.71 (+1.01)**|**72.90 (+1.16)**|**72.81 (+1.20)**|**72.78 (+1.29)**| **72.71 (+1.28)**|
>
> ***Table B**: Accuracy drop comparison when distilling ResNet18 from different teacher models (ResNet34 vs. ViT-L).*
>
> |Method/Teacher|ResNet34|ViT-L (Acc. Drop)
> |:-:|:-:|:-:|
> |Teacher Acc.|73.31|85.14|
> |Student (ResNet18)|69.75|69.75|
> |DKD|71.70|-0.27 (71.43)|
> |DKD + GPD|72.71|**-0.00** (72.71)|
>
>
> **Q3. How can we measure the performance gap on knowledge transfer? It would be helpful to provide a clearer idea of the best range for this gap.**
>
> We agree that it would be very useful for the distillation community if the best range of the performance gap can be accurately provided. However, it is non-trivial in practice since this gap may vary across different datasets and tasks. For example, a performance gap of 3\% or 5\% could be good for training classification models on ImageNet. But this gap will definitely not be a suitable value for training on MNIST since almost all the models have an accuracy over 98\%. More critically, if we consider other tasks, e.g. image restoration, accuracy cannot be directly used to measure the gap.
>
> Instead of seeking a universal "best gap", we propose to cast the problem of determining the best gap into building a suitable gap in terms of model size between the student and the teacher. Specifically, we use channel expansion ratio $r$ and the number of branches $M$ as a proxy measure to control this gap. As shown in Figure 4, configurations with $r > 3$ and $M > 6$ typically lead to a "too-large performance gap", such that a too-strong teacher hampers knowledge transfer. In contrast, a small expansion ratio (e.g., 2 or 3) combined with $M = 6$ brings a significant performance improvement of at least 1\% on ImageNet. In practice, we recommend using $M=6$ and $r=2$ and this setting **generalizes well to diverse scenarios**. As demonstrated in Section 4, these configurations consistently yield significant improvements across various architectures and distillation scenarios. Based on the above, **building the dynamic teacher with $M=6$ and $r=2$ can be used as a good gap**.

---

> ### Author Response · Authors · 2024-11-23
> **Authors' Response to Reviewer 49EM (Part 3)**
>
> **Q4. Overengineering in the Dynamic Teacher Model. The necessity of the complex structure designed in this paper, including Inverse Reparameterization (IR) and parameter sharing, appears somewhat disconnected from the core problem. If the primary goal is to reduce the performance gap, it remains unclear whether the proposed architecture (specifically the IR and shared parameters) is required to achieve this. Making the connection between these design choices and the main goal a bit clearer could help the paper feel more cohesive. Has the author considered more straightforward options to close the gap? If this complex setup is essential, could they explain more about how it directly addresses the problem?**
>
> We highlight that the proposed designs are necessary to control the performance gap between student and teacher. Indeed, there exists a straightforward approach that uses a randomly initialized dynamic teacher without either Inverse Reparameterization (IR) or parameter sharing. In fact, it works well for training models from scratch, but it fails in fine-tuning scenarios since the randomly initialized teacher can easily destroy the distillation of a pretrained student model. This motivates our design of IR, which initializes the dynamic teacher to start with the same accuracy as the student, making our method effective for both training from scratch and fine-tuning settings.
>
> On the other hand, the parameter sharing mechanism is also important since it acts as a **hard constraint** to control the gap between student and dynamic teacher within a suitable range. In this way, it is hard for the dynamic teacher to become significantly better than the student since they share/optimize the same set of parameters. Interestingly, parameter sharing also benefits the training of the student since the gradients come from optimizing both the dynamic teacher and the student. In fact, this has been empirically verified. Our ablation study (Table 4) demonstrates that parameter sharing contributes an additional 0.35% accuracy improvement. Together with IR, our GPD is able to effectively maintain appropriate performance gaps throughout the entire training process. We have revised the paper to better highlight the necessity and clarify the connection between these design choices and the primary goal.

---

### Author Response · Authors · 2024-11-23
**# Response to all reviewers and area chairs**

Dear Reviewers and Area Chair,

We thank all reviewers and area chairs for their insightful comments and valuable time. We are grateful to all reviewers for reaching agreement on the following aspects:

- **The Novel Approach**:
  - "The proposed approach is **novel**. It co-trains a dynamic teacher to alleviate the teacher-student capacity gap and applies inverse reparameterization and channel-branch reparameterization for efficient parameter sharing." (Reviewer J6Y5)

- **The Sound Methodology**:
  - "The problem is **well-motivated**, particularly in lines 54-61, highlighting the importance of adaptively managing the performance gap." (Reviewer J6Y5)
  - "The problem of effectively performing KD with an extremely large teacher model is important and **well-motivated**." (Reviewer FcyL)

- **Well-Written and Organized**:
  - "The paper is **well-organized** and **easy to follow**, with **clear descriptions**, **intuitive figures**, and **rigorously detailed algorithms**." (Reviewer J6Y5)
  - "The paper is **well-organized** and **easy to follow**." (Reviewer FcyL)

- **Comprehensive and Effective Experimental Evaluation**:
  - "...the method **effectively narrows this gap**, facilitating more efficient knowledge transfer." (Reviewer 49EM)
  - "The proposed GPD demonstrates **adaptability across various training setting**s, with **significant improvements** across multiple models and datasets. **Extensive ablation studies** further validate the method’s effectiveness." (Reviewer 49EM)

We have taken each comment very seriously and put tremendous effort into addressing them. Moreover, we also carefully revised the paper. Here, we offer a summary:

- We provide **theoretical analysis** and **empirical evidence** explaining why large performance gaps hinder knowledge distillation.
- We demonstrate that a performance gap of approximately 3% (achieved with expansion ratio r=3 and branch number M=6) is optimal for effective knowledge transfer.
- We provide extensive comparisons with **teacher-assistant methods** and **self-distillation approaches**, demonstrating that GPD **consistently outperforms these methods**.
- We clarify that re-parameterization incurs **negligible cost** (0.021 ms per operation).
- We provide the training costs, showing that our GPD introduces only 33\% additional overhead, while achieving a significant 1.58% performance improvement. Our GPD offers a **favorable trade-off** between performance and training cost.
- We demonstrate **robust performance across different expansion ratios** without task-specific tuning.
- We highlight **the necessity of IR and parameter sharing** to reduce the performance gap, with IR enabling generalization and parameter sharing establishing explicit constraints.
- We analyze GPD’s **effectiveness** in mitigating extreme teacher-student gaps.
- We clarify that the added noise leads to a **negligible accuracy difference** of approximately 0.002%.
- We provide mathematical formulation and illustrations showing how our re-parameterization method **naturally extends to transformer**.

Thanks again to all reviewers and area chairs. We are eager to have a fruitful discussion with all the reviewers and we look forward to any comments, questions, and suggestions.

Best Regards,

Authors

---

### Author Response · Authors · 2024-11-26
**Thanks to all reviewers and area chairs**

Dear Reviewers and Area Chairs,

We sincerely thank all reviewers for their constructive feedback and acknowledgment on our work. We are highly encouraged by the positive consensus reached by all the reviewers: Reviewer 49EM initially providing a positive assessment (score of 6), Reviewer FcyL raising the score to 6, and Reviewer J6Y5 promising to raise the score to 6 after reading our rebuttal. We have carefully revised the paper according to the concerns from all the reviewers.

We are convinced that the idea of preserving a suitable performance gap to enhance the transfer of knowledge is novel and particularly effective. We believe that our work will serve as a strong baseline for both knowledge distillation and model initialization. We will release our code to facilitate the reproducibility and future research.

We appreciate your time and consideration throughout this review process.

Best regards,

Authors of #869

---

### Meta-Review · Area_Chair_QK7T · 2024-12-17

**Metareview:**

After discussion, the major concerns about novelty, contribution and experiments of the proposed approach were mostly solved. The main contribution lies in a dynamic teacher approach to alleviate the teacher-student capacity gap and applies inverse re-parameterization and channel-branch re-parameterization for efficient parameter sharing. The paper defines an improved teacher-assistant method for the important self-distillation problem.

As pointed out by the reviewer, there remains a large performance gap between the teacher and student models after applying the proposed distillation approach, which is not fully explained and not well solved after rebuttal. As shown in Table 10 in the revised paper, the performance improvement upon the ImageNet benchmark over a 2022-year baseline is marginal (72.71 vs. 72.60).

**Additional Comments On Reviewer Discussion:**

After discussion, all the reviewers gave the positive scores. Reviewer 49EM initially gave the positive score. After rebuttal, Reviewer FcyL raised the score to 6, and Reviewer J6Y5 updated the score from 5 to 6. The authors carefully responded to the questions and revised the paper according to the concerns from all the reviewers.

---

### Decision · Program_Chairs · 2025-01-22

Accept (Poster)